# Activation of cytomegalovirus-encoded G protein-coupled receptor UL33 by an innate N-terminal peptide
Anna K. Drzazga [1], Shota Suzuki [2], Caroline Wouters [1,3], Felix Faas [1], Kouki Nishikawa[4], Akiko Kamegawa[2], Yoshinori Fujiyoshi [2], Mette M. Rosenkilde [1] ✉ & Naotaka Tsutsumi [2] ✉

Human cytomegalovirus (HCMV) encodes the orphan G protein-coupled receptor (GPCR) UL33, which exhibits constitutive activity that disrupts host G protein signalling, facilitating efficient viral replication and pathogenesis. The cryo-electron microscopy (cryo-EM) structure of UL33 bound to the $G_s$ subtype of G protein reveals the N-terminal peptide as a tethered ligand reminiscent of the protease-activated receptors and adhesion GPCRs. This self-agonism induces a non-canonical active state that facilitates promiscuous G protein coupling, a plausible viral strategy for fine-tuning host signalling. Structure-guided mutagenesis disrupting key interactions between the N-terminus and its binding pocket abolishes G protein-mediated signalling, confirming the role of the N-terminus as a self-agonist. Our findings elucidate the structural basis for this activation mechanism and highlight the strategies employed by HCMV to hijack host G protein signalling.

UL33 is a G protein-coupled receptor (GPCR) encoded by human cytomegalovirus (human CMV or HCMV)[1]. It plays key roles in the viral lifecycle by promiscuously activating the $G_s$, $G_i$ and $G_q$ subtypes of G proteins in a ligand-independent manner, thereby disrupting host signalling[2-5] (Fig. 1A). However despite its importance the molecular mechanism underlying its activation has remained elusive. This is largely because UL33 is an 'orphan' receptor with no identified ligand even though it is considered an orthologue of a host CC chemokine receptor (CCR)[6].

HCMV is a highly prevalent β-herpesvirus that establishes lifelong latency infecting approximately 60% of adults in developed countries and over 90% in developing countries[7]. While often asymptomatic HCMV can cause various diseases. Congenital HCMV infection, affecting ~1% of newborns often leads to moderate to severe outcomes, such as hepatitis, nephritis, and colitis[8]. These complications can result in long-term effects like developmental delays and defects including hearing loss, vision impairment, and intellectual disabilities[9]. In the context of organ transplantation HCMV infection can trigger severe inflammatory responses and research is currently underway to eliminate HCMV from transplanted organs[10,11]. Moreover HCMV infection has been linked to an increased risk of various diseases, including certain types of cancer[12].

UL33 is one of four virally encoded GPCRs (vGPCRs) that CMV has captured from its hosts throughout evolution[1,13,14]. These vGPCRs fall into two distinct groups based on their evolutionary history. The first, comprising US27 and US28, is found only in CMVs that infect humans, apes, and Old World monkeys. Their ancestral receptor has been identified as the chemokine receptor $CX_3CR1$ (ref. 15) and US28 maintains the binding capacity to the $CX_3CR1$ ligand $CX_3CL1$ with extended promiscuity[16,17]. Consequently the molecular functions and three-dimensional structures of this family are relatively well-characterised[6]. On the other hand UL33 and UL78 are more primordial vGPCRs, acquired by CMVs infecting rodents at much earlier evolutionary stages[18,19] and are thus expected to have important evolutionary functions in the CMV lifecycle[20]. However their amino acid sequences appear highly divergent from the original host GPCRs, making them a particularly challenging and enigmatic group to study. In particular, the 'orphan' status of UL33 has shifted the research focus toward its ligand-independent, constitutive activities[6] (Fig. 1A). For example UL33 drives the activation of the cAMP response element (CRE)-binding protein (CREB) transcription factor, enhancing its recruitment to the major immediate early locus to promote viral reactivation[4]. Furthermore a recent study demonstrated that UL33 robustly stimulates $G\alpha_q$-dependent PLC-β signalling for efficient lytic replication in epithelial cells[5]. Collectively these findings establish UL33 as a key factor in the CMV lifecycle. However it has remained the only HCMV GPCR without an experimental structure[21,22], which has hampered a mechanistic understanding of its activation.

[1]Department of Biomedical Sciences, Laboratory for Molecular and Translational Pharmacology, University of Copenhagen, Copenhagen, Denmark. [2]Advanced Research Initiative, Institute of Integrated Research, Institute of Science Tokyo, Bunkyo-ku, Tokyo, Japan. [3]Synklino A/S, Copenhagen, Denmark. [4]Joint Research Course for Advanced Biomolecular Characterization, Faculty of Agriculture, Tokyo University of Agriculture and Technology, Fuchu, Tokyo, Japan. ✉e-mail: rosenkilde@sund.ku.dk; tsutsumi.naotaka@tmd.ac.jp

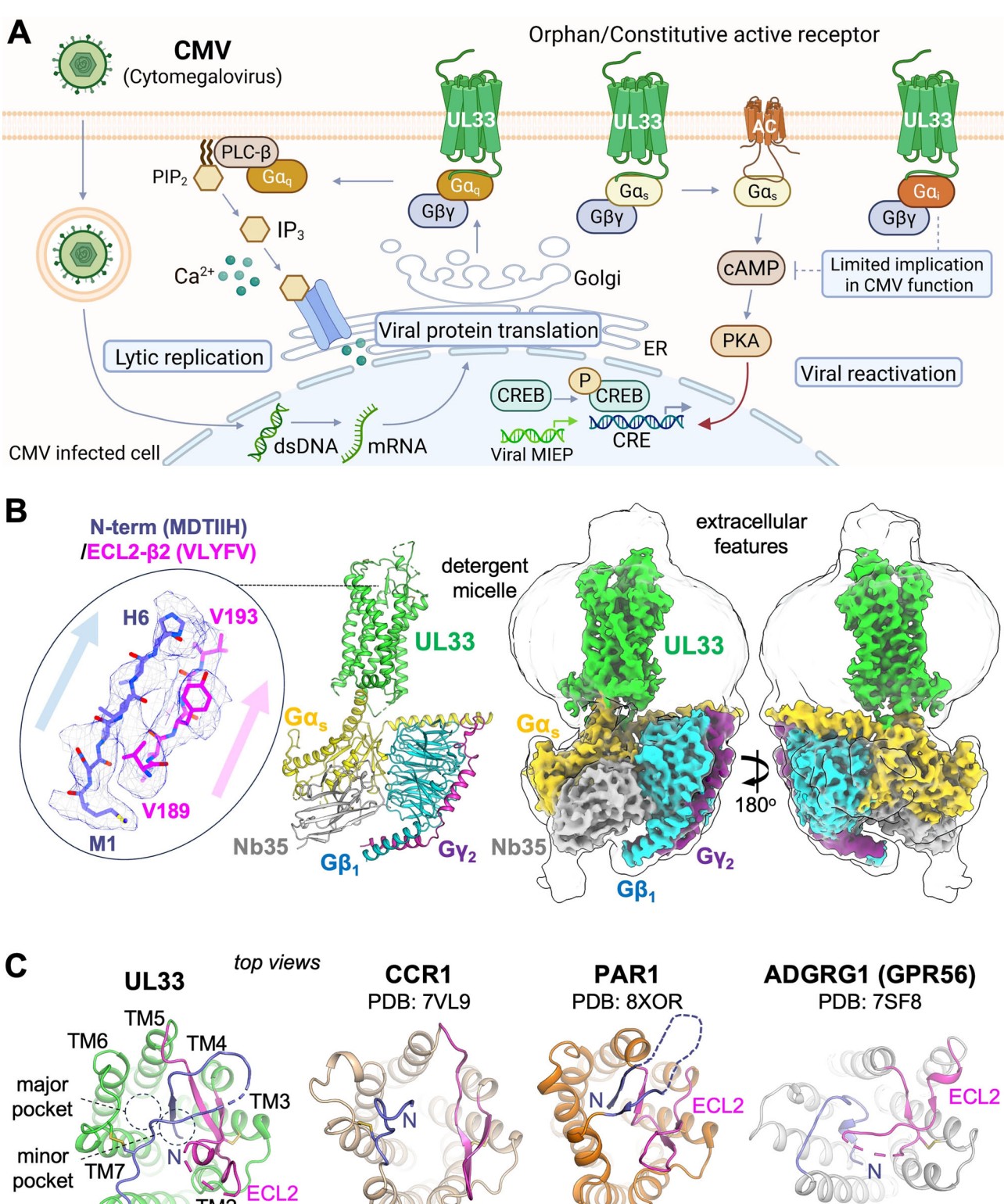

Here we combined structural analysis with receptor mutagenesis in cell signalling assays to reveal the activation mechanism of UL33. The cryo-electron microscopy (cryo-EM) structure of the UL33-$G_s$ complex showed that the N-terminal loop inserts into UL33's extracellular pocket, reminiscent of activation in protease-activated receptors (PARs, class A) and adhesion GPCRs (aGPCRs, class B2)[23,24]. This innate machinery stabilises a non-canonical active state to fine-tune promiscuous G protein signalling

possibly as a viral pathogenic strategy. Mutational disruption of key interactions between the N-terminal peptide and peptide-recognition pocket abolished $G_s$-mediated signalling while maintaining surface and total receptor expression, demonstrating that it serves as a tethered agonist for UL33. Furthermore we observed a narrow cavity connecting to the empty major ligand-binding pocket adjacent to the N-terminal self-agonist, offering a potential target for allosteric modulators. As N-terminal

**Fig. 1 | Cryo-EM structure of the UL33-G$_s$-Nb35 complex. A** Schematic representation of the UL33 signalling and its role in infected cells during the CMV lifecycle. After CMV enters the host cell the viral genome is transported to the nucleus, where its double-stranded DNA (dsDNA) is transcribed into messenger RNA (mRNA) by the host cell's transcriptional machinery. The viral mRNA is then translated into proteins in the cytoplasm and viral membrane and secreted proteins are directed to the Golgi apparatus for processing, including glycosylation. The UL33 protein is expressed at the plasma membrane and functions as an orphan constitutively active receptor that signals through the host Gα$_s$, Gα$_i$, and Gα$_q$ pathways. UL33 plays a key role in activating CREB through Gα$_s$ signalling via adenylate cyclase (AC) activation, which promotes viral reactivation, while UL33 also stimulates Gα$_q$-dependent PLC-β signalling, a pathway critical for efficient lytic replication. Additional abbreviations in the figure: PLC-β phospholipase C-beta, PIP$_2$ phosphatidylinositol 4,5-bisphosphate, IP$_3$ inositol trisphosphate, ER endoplasmic reticulum, P phosphate, MIEP major immediate early promoter. **B** A ribbon model and a cryo-EM Coulomb potential map of the HCMV UL33-human G$_s$-Nb35 complex. A stick model of UL33's parallel β-sheet between the N-terminal six amino acids (MDTIIH, blue carbon) and ECL2-β2 (VLYFV, magenta carbon) is displayed with the cryo-EM map (contour level 6σ, sharpening B factor −30 Å$^2$). The ribbon model and sectioned map of the overall views are coloured green (UL33), yellow (Gα$_s$), cyan (Gβ$_1$), purple (Gγ$_2$), and grey (Nb35). For the sectioned sharpened cryo-EM map (contour level 6σ), the unsharpened and smoothed map is overlaid with the transparent grey colour to show the unmodelled features at the lower threshold. **C** Extracellular structural comparison among UL33 (green), CCR1 (wheat, PDB: 7VL9), PAR1 (orange, PDB: 8XOR), and ADGRG1 (grey, PDB: 7SF8) in G protein-bound active states with extracellular loop 2 (ECL2) and receptor N-termini shown magenta and blue, respectively. G proteins and other subunits in the complexes are not displayed for clarity.

## Table 1 | Cryo-EM data collection, refinement and validation statistics

| | UL33-G$_s$-Nb35 (EMD-65918) (PDB 9WEY) |
|---|---|
| **Data collection and processing** | |
| Magnification | 63,291x |
| Voltage (kV) | 300 |
| Electron exposure (e⁻/Å$^2$) | 50 |
| Defocus range (μm) | −0.6 to −1.2 |
| Pixel size (Å) | 0.79 |
| Symmetry imposed | C1 |
| Initial particle images (no.) | 1,502,019 |
| Final particle images (no.) | 54,507 |
| Map resolution (Å) | 3.3 |
| FSC threshold | 0.143 |
| Map resolution range (Å) | 2.5–5.5 |
| **Refinement** | |
| Initial model used (PDB code) | ColabFold (UL33)/PDB: 7XT8 (G$_s$-Nb35) |
| Model resolution (Å) | 3.7 |
| FSC threshold | 0.5 |
| Model resolution range (Å) | N/D |
| Map sharpening B factor (Å$^2$) | −30 |
| **Model composition** | |
| Non-hydrogen atoms | 8084 |
| Protein residues | 1046 |
| Ligands | N/A |
| **B factors (Å$^2$)** | |
| Protein | 106.37 |
| Ligand | N/A |
| **R.m.s. deviations** | |
| Bond lengths (Å) | 0.003 |
| Bond angles (°) | 0.662 |
| **Validation** | |
| MolProbity score | 1.76 |
| Clashscore | 9.07 |
| Poor rotamers (%) | 0 |
| **Ramachandran plot** | |
| Favoured (%) | 95.98 |
| Allowed (%) | 4.02 |
| Disallowed (%) | 0 |

*N/D not determined, N/A not applicable.*

modifications are critical for the function of the UL33 family of receptors[18,25], the mechanistic data presented here provide a roadmap for developing agents that modulate pathogenic UL33 activity and fill the gap in our knowledge of the HCMV 'GPCRome', which collectively manipulates 'hijacked' host cells[6].

## Results

### Cryo-EM structure reveals a tethered N-terminal self-agonist

To obtain stable UL33-G protein complexes we employed the NanoBiT tethering strategy[26], where the LgBiT and HiBiT sequences[27] were fused at the C-terminus of full-length UL33 and Gβ$_1$, respectively. Additionally the Protein C epitope was placed between UL33 and the LgBiT tag to facilitate affinity purification. UL33 and heterotrimeric G protein were co-expressed in the Expi293 system with doxycycline-induced receptor expression. We successfully purified UL33 bound to G$_s$, G$_i$ and G$_q$ by anti-Protein C affinity chromatography followed by size-exclusion chromatography (SEC) (Supplementary Fig. 1A top), indirectly supporting the notion that UL33 can activate the three subtypes of G proteins[3] as non-productive Gα subunits can dissociate from the complex even with the Gβ tethering[22].

We then performed single-particle cryo-EM analysis with the aid of G protein stabilising chaperones Nb35 for G$_s$ and scFv16 for G$_i$, and 'G$_q$' where the N-terminal region of Gα$_q$ was replaced by the scFv16-binding G$_i$ sequence[28,29]. However only the UL33-G$_s$-Nb35 complex yielded an interpretable 3D reconstruction, indicating that the interactions between UL33 and G$_i$ or G$_q$ are structurally flexible (Supplementary Fig. 1A–D). Focusing on the G$_s$ complex for 3D cryo-EM analysis we obtained a 3.3 Å nominal resolution map, despite a variable local resolution with a better-defined G protein density, suggesting that the UL33-G$_s$ interface is also tenuous (Fig. 1B; Table 1; Supplementary Figs. 2 and 3).

The core region of UL33 exhibits the canonical 7-transmembrane helix (TM) GPCR fold like other vGPCRs and the putative ancestral chemokine receptor, CCR1 (refs. 6,30) (Fig. 1B and Supplementary Fig. 4). The most notable feature is the insertion of an N-terminal β-strand into the extracellular GPCR pocket to form a parallel β-sheet with the second β-strand of the extracellular loop (ECL) 2 (Fig. 1B), suggesting that it functions as the tethered 'self-agonist', consistent with UL33's constitutive G protein activities. It should be noted that the cryo-EM density for this N-terminal peptide is of relatively poor quality (Fig. 1B left inset; Supplementary Fig. 3D) implying a high degree of local flexibility or dynamics. Nevertheless similar extracellular occlusions were observed for other vGPCRs, such as Epstein–Barr virus (EBV)-encoded BILF1 and HCMV-US27 but mainly with ECL2 instead of the N-terminus (Supplementary Fig. 4A) for constitutive activation and inactivation, respectively[21,31]. This N-terminal tethered agonism mechanism which we previously predicted[6], is reminiscent of PARs and aGPCRs (Fig. 1C), where the cleavage-exposed N-terminal loops act as agonists for receptor activation[23,24]. PAR1 is more analogous to UL33 in that its N-terminal region forms a β-strand[21,32]. However unlike cleavage-activated GPCRs, which require proteolysis to expose their agonist peptide,

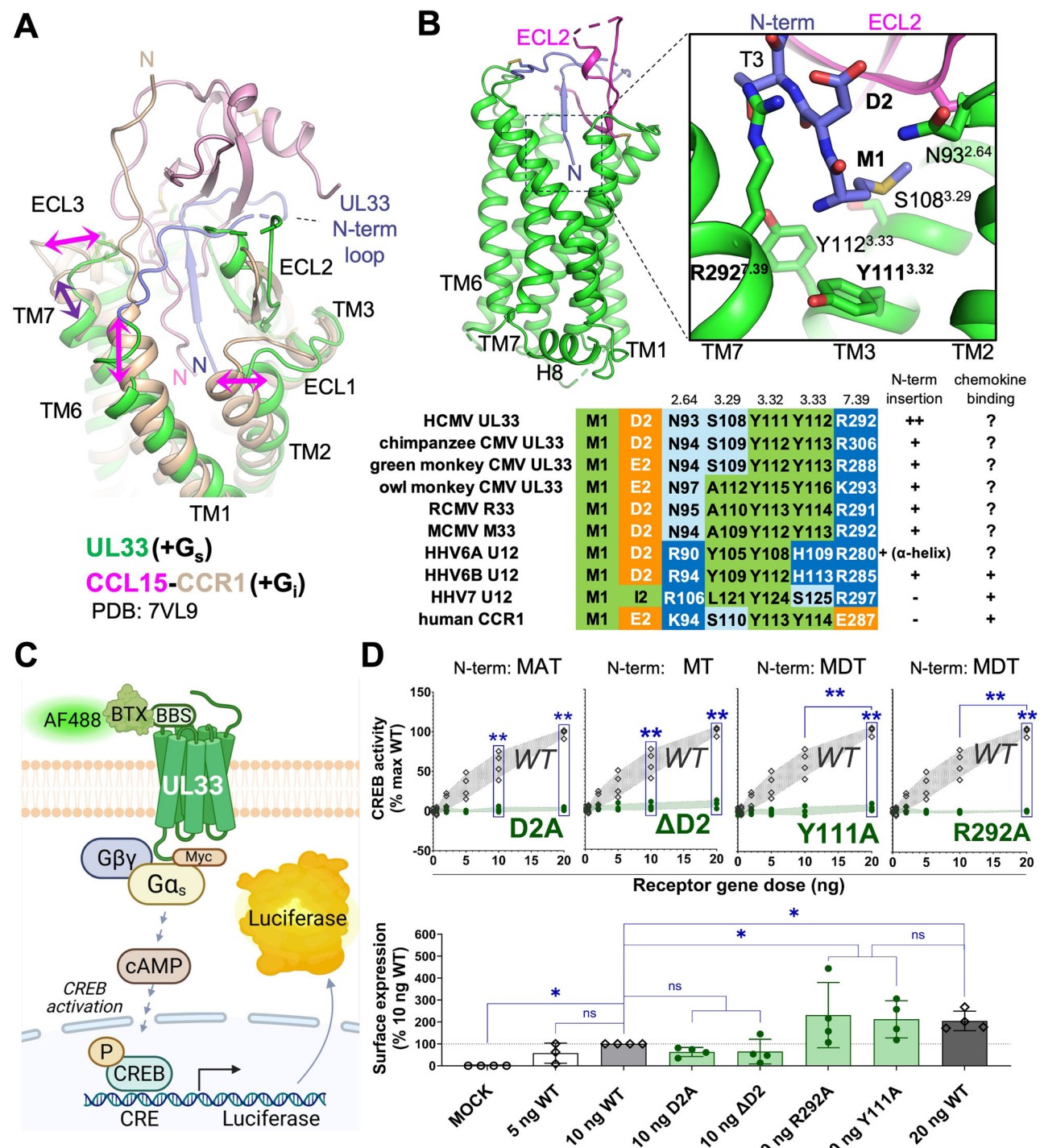

**Communications Biology** | (2026)9:415

UL33 utilises its nascent N-terminus, with the first methionine playing a central role.

## N-terminal peptide occludes the receptor pocket as a chemokine mimic

The tethered peptide forms a three-stranded β-sheet together with ECL2 and extends into the 'minor' ligand-binding pocket[33] of UL33 while the 'major (main)' pocket remains empty (Fig. 1C and Supplementary Fig. 4A, B). Consequently the extracellular cavity is narrowed compared to CCR1, which limits the entry of larger ligands, such as chemokines (Fig. 2A). This unique extracellular architecture also results in the occlusion of the defined chemokine recognition sites (CRSs) CRS1, CRS1.5, CRS2 and CRS3,

preventing chemokine binding (Supplementary Fig. 4C)[30,34]. If the entire N-terminal loop acts as the 'chemokine mimetic' the N-terminal tip functions at CRS2, which is considered critical for receptor activation, while the first β-strand and flexible loop connecting to TM1 are located at CRS3, pushing ECL2 (Fig. 2A). In the human chemokine system such as the CCL15-CCR1 complex, the receptor's N-terminal loop is exposed to the extracellular space to capture chemokine at CRS1 and the chemokine body often pushes ECL3 towards CRS1.5 in addition to the CRS2 and CRS3 contacts[30]. Accordingly UL33's unique N-terminal structure reorganises the extracellular architecture in comparison with the CCL15-CCR1 system, where TM1 and TM2 bend away from each other, with the ECL3 moving inside to narrow the extracellular pocket (Fig. 2A).

**Fig. 2 | The N-terminal peptide works as a self-agonist for UL33. A** The UL33 N-terminal loop partly mimics chemokine binding to human chemokine receptors. Superimposition of UL33 (green) and CCL15-CCR1 (pink and wheat, PDB: 7VL9). The UL33 N-terminal loop (residue 1–23) is coloured blue. The magenta double arrows indicate a positional difference of corresponding TMs between UL33 and CCR1, and the purple double arrow shows the relative extracellular length of TM7. **B** Interaction of the N-terminal M1D2 motif within UL33's extracellular pocket. The overall side view of UL33 is 90 degrees rotated relative to (**A**) and includes a window to show the magnified region. The key amino acid and corresponding residues of the HCMV UL33 orthologues are compared at the bottom. Chimpanzee (Ape) CMV, PaHV2; green monkey (Old World monkey) CMV, CeHV5; owl monkey (New World monkey) CMV, AoHV1; HHV, human herpesvirus. The numbering (x.xx) on top of the sequences indicates the Ballesteros–Weinstein numbering of class A GPCRs. The colour scheme is as follows: acidic, orange; basic, blue; aliphatic and aromatic without charge, green; polar, light blue. The presence or absence of the N-terminal insertion is based on AlphaFold3 (https://alphafoldserver.com/) where − indicates no prediction of the N-terminal insertion, + indicates prediction with low confidence ($70 > \text{pLDDT} \geq 50$), and ++ indicates prediction with high confidence ($\text{pLDDT} \geq 70$). The chemokine binding ability is based on the literature[2–5,18,20,25,34,37–42]. **C** The principle of the assays applied to study the expression and constitutive activity of UL33 variants. Receptor expression in the transfected HEK293A cells was detected by labelling the α-bungarotoxin binding site (BBS) implemented in ECL2 of the receptor with Alexa Fluor 488 (AF488)-conjugated α-bungarotoxin (BTX), and subsequent fluorescence measurements. UL33-mediated $G_s$ signalling was measured via bioluminescence readout utilising the trans-reporting CREB system, leading to luciferase synthesis upon CREB activation. **D** The

impact of the amino acid residues maintaining N-terminal insertion in UL33 on its activity and expression in HEK293A cells. CREB signalling activity was verified for the following UL33 mutants: D2A, ΔD2, Y111A, and R292A (represented by green circles in the individual plots) and compared with the wild type activity (WT, grey open squares), across receptor gene doses ranging from 0 to 20 ng of DNA per 35,000 cells. The signalling results were obtained from $n = 4$ biologically independent experiments each performed in technical triplicate. Data were normalised to mock transfection (0% signal) and the highest gene dose of the WT UL33 (100% signal). Results are presented as individual data points for each experiment superimposed on a shaded area depicting the standard deviation (grey for WT and light green for mutants). The surface receptor expression data were also obtained from $n = 4$ biologically independent experiments each performed in technical triplicate, using 5, 10 and 20 ng gene doses for WT and a 10 ng gene dose for the mutants per 35,000 cells. The expression results are presented individually normalised to mock transfection (0%) and 10 ng of WT DNA per 35,000 cells (100%). Data represent mean ± standard deviations. The differences in CREB activity between each of the UL33 mutants and WT at the highest gene dose as well as between gene doses ensuring similar surface expression levels of WT UL33 and its mutants, were determined using two-way ANOVA with Fisher's LSD post hoc test. Significant differences in expression levels between gene doses of WT UL33 and its mutants were determined using one-way ANOVA and Fisher's LSD post hoc test. Data sets compared between WT and a mutant are marked with an open rectangle when originating from the same gene dose and with a bracket when the gene doses of WT and the mutant are different. In all cases statistical significance is indicated as follows: *$P < 0.05$, **$P < 0.01$, ns–not significant.

At the mimicked 'CRS2', UL33's first methionine residue reaches the deepest part of the minor pocket and the M1 sidechain primarily interacts with $S108^{3.29}$, $Y111^{3.32}$, and $Y112^{3.33}$ at TM3, while the second residue, aspartate (D2), projects its sidechain to the opposite side of the M1 sidechain, forming electrostatic interactions with $R292^{7.39}$ at TM7 and $N93^{2.64}$ at TM2 (Fig. 2B top) (superscript refers to Ballesteros–Weinstein numbering). This 'M1D2' motif forms a bridge between TM3 and TM7 which are typically important for class A GPCR activation[35]. The key residues including the broader N-terminal region, are well conserved in vGPCRs closely related to UL33, such as M33 and R33 encoded by mouse CMV (MCMV) and rat CMV (RCMV), respectively, as well as UL33 encoded by CMV infecting Old World monkeys (e.g. green monkey) and New World monkeys (e.g. owl monkey) (Fig. 2B bottom; Supplementary Fig. 5), suggesting a common mechanism of activation. AlphaFold3 (ref. 36) (https://alphafoldserver.com/) generated models for many related receptors that featured an N-terminal insertion, although the confidence for these insertions was low, as indicated by their low predicted local distance difference test (pLDDT) scores ($70 > \text{pLDDT} \geq 50$). Despite this low confidence we note that the presence or absence of the predicted N-terminal insertion is generally correlated with that of identified chemokine ligands[2–5,18,20,25,34,37–41]. While HHV6B U12, a functional CCL5 receptor[42], is a notable exception, this potentially indicates that the N-terminal architecture is even more flexible for HHV6B U12 compared to that of UL33, allowing the extracellular cavity to open and accommodate chemokine binding. These observations also highlight the current limitations of in silico prediction for such flexible regions and underscore the importance of the experimental structure we have determined.

**N-terminal tethered agonist is indispensable for UL33 activation**

To probe the importance of the critical amino acid residues maintaining the N-terminal insertion we generated three alanine mutants and one deletion mutant that disrupt this unique architecture. We then tested these UL33 variants for $G_s$ signalling by measuring CREB activities (Fig. 2C, D and Supplementary Fig. 6), an assay previously used for the UL33 system[3].

Consistent with the previous studies, expression of UL33 induced robust gene-dose-dependent activation of CREB, although UL33 is coupled to both $G_s$ and $G_i$, which increases and decreases CREB activity, respectively (Supplementary Fig. 6A, B left). This indicates that UL33-mediated signalling is predominantly $G_s$-coupled in human embryonic kidney (HEK)

293 cells with a minor $G_i$ component. Therefore we used CREB activity as a proxy for UL33-$G_s$ signalling. Furthermore to facilitate robust monitoring of cell surface and total UL33 expression without modifying the receptor N-terminus, we engineered UL33 to insert the α-bungarotoxin (BTX) binding site (BBS) tag[43] within ECL2 guided by the cryo-EM structure, together with an additional C-terminal Myc tag for supplementary detection. To confirm that these modifications did not affect function this double-tagged construct was assessed for its activity in CREB signalling together with the single-tagged BBS-UL33, UL33-Myc and the untagged version. All four constructs exhibited comparable dose-dependent signalling (Supplementary Fig. 6A, B) and the double-tagged BBS-UL33-Myc was selected as a template for creating the UL33 mutants for subsequent studies.

Remarkably all three alanine substitutions, D2A, Y111A, and R292A, as well as the D2 deletion (ΔD2), completely abrogated UL33 signalling without impairing receptor expression (Fig. 2D; Supplementary Figs. 6C, D and 7), confirming the hypothesis that the N-terminal peptide mainly functions as a self-agonist, rather than being merely required for receptor stability and expression. Indeed detailed expression analysis revealed the mutants have distinct molecular fates (Fig. 2D bottom; Supplementary Fig. 6C). The Y111A mutant showed a two-fold increase in surface expression with unchanged total expression compared to WT indicating a redistribution to the plasma membrane, possibly due to inhibited constitutive internalisation. Conversely the R292A mutation increased both surface and total expression two-fold, suggesting enhanced protein production or improved protein stability by preventing signal-induced degradation. Despite these divergent effects on receptor trafficking and stability and the minor changes for D2A and ΔD2, their shared phenotype, the complete abrogation of signalling, confirms the critical functional role of these residues.

**Functional impact of N-terminal variations in the UL33 family**

The N-terminal deletion and extension (Fig. 3A) also affect the UL33 function. UL33S, a natural translation variant with a 22-amino-acid N-terminal deletion from the canonical UL33[40], resulted in the loss of cell surface expression and signalling[3] while the total expression appeared increased[5]. Similarly the N-terminal extremity is required for MCMV M33, as the deletion of 10 amino acid residues and addition of HA or Myc tags at the N-terminus abolished surface expression and signalling[18,25], reinforcing the existence of the common architecture and receptor activation

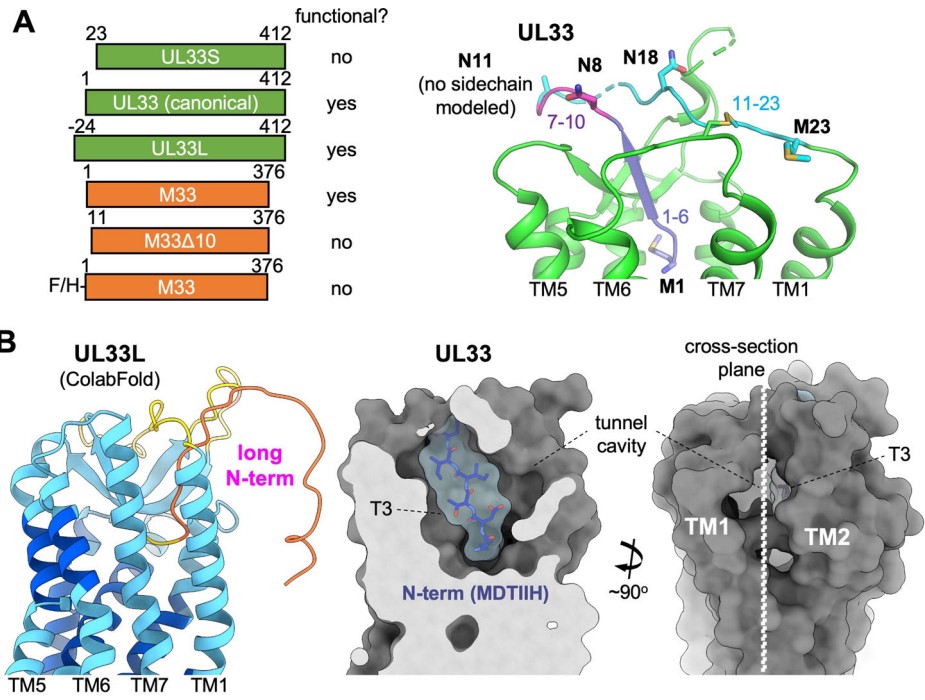

**Fig. 3 | Deletion and extension of the UL33 N-terminus. A** The N-terminal modified UL33. The left panel shows the linear schematic of the natural and synthetic N-terminal variants of UL33 and M33 with information about their functionalities and the right panel shows the structure of the N-terminal region in the UL33 extracellular architecture with segmented colouring (blue, residues 1–6; purple, residues 7–10; cyan, residues 11–22). The first and second methionine residues and asparagine residues in the consensus N-glycosylation site are displayed as thick sticks. **B** The ColabFold-predicted UL33L structure displaying a cross-sectional view of the UL33 structure, which shows the tunnel cavity connecting to the N-terminal tip. The ColabFold structure is shown as a ribbon model at the left panel and coloured by the prediction score (blue, pLDDT ≥ 90; light blue, 90 > pLDDT ≥ 70; yellow, 70 > pLDDT ≥ 50; orange, 50 > pLDDT). The cross-section view of UL33 is displayed on the centre and coloured dark and light grey to indicate the surface and cross-section of the protein, respectively, while the N-terminal six amino acid residues are shown as blue sticks and a semi-transparent surface. The cross-section plane (white dashed line) is shown with the UL33 surface model on the right viewed from an angle that peeks into the cavity tunnel.

mechanism. However the N-terminal extension appears to affect HCMV UL33 differently, as N-terminal FLAG and HA tags have been used to monitor UL33 expression and facilitate receptor pull-down[44,45]. Furthermore an additional UL33 variant, termed UL33L, was reported recently in a preprint, which has a 24-amino-acid N-terminal extension (MLAFLQN-PEASSPRTAPAVCYSPT) from the canonical UL33 N-terminus. This variant appeared to be as active as UL33 in $G_q$/PLC-β signalling raising questions about the role of the extra residues[5]. With the experimental structure as a template ColabFold structure prediction[46] demonstrated that this protrusion can turn and escape from the narrow orthosteric cavity without reaching the major ligand-binding pocket (Fig. 3B).

These observations are particularly interesting as the *UL33* coding gene is interrupted by an intron, separating the exons that encode the first eight amino acid residues (MDTIIHNS) and the rest of the receptor. This gene structure is analogous to that of related vGPCRs, MCMV UL33 orthologue, M33 and U12s encoded by related human β-herpesviruses HHV6/7 (ref. 40) (Supplementary Fig. 8). In contrast CCR1, a putative ancestor of UL33, consists of a single coding exon[47]. Furthermore many herpesvirus genes are intronless even when the original host gene contains this segment, a result of capturing intron-free cDNA copies of host mRNAs[48]. These facts suggest that the shared two-exon structure of the UL33 gene family likely evolved before the divergence of *Betaherpesvirinae* into the *Cytomegalovirus* and *Roseolovirus* genera. Subsequently within the CMV lineage, this N-terminal region appears to have been adapted for self-activation.

## UL33 adopts a non-canonical active state with an inwardly retained TM6

In its active G protein-bound state the overall 7TM fold of UL33 is similar to those of CCR1 and CX₃CR1, with the Cα root-mean-square deviations (RMSDs) of 3.09 and 2.85 Å, respectively. However at the intracellular ends, UL33's TM5 and TM7 are positioned 5 and 4 Å outward, respectively, while TM6 is inwardly retained, located 6 Å closer to the central axis compared to CCR1 (Fig. 4A). This structural arrangement contrasts with that of most other activated CCRs and typical class A GPCRs which exhibit a canonical agonist-induced outward rearrangement of TM6[35]. This canonical movement is exemplified by CCR1[34] (Fig. 4A) and CCR5, in which the

intracellular end of TM6 is displaced outward by 10 Å upon agonist binding compared to its inverse agonist-bound state[49].

The TM positions are more analogous between UL33 and $G_i$-bound CX₃CR1[50] (Fig. 4A) and the small TM6 movement was also observed for the PAR1-$G_i$/$G_q$ complexes activated by the nascent N-terminus after thrombin cleavage[23], highlighting the diversity of the G protein recognition mechanism. By comparing the DR$^{3.50}$Y and NP$^{7.50}$xxY motifs, UL33-R129$^{3.50}$ faces downward to interact with V246$^{6.37}$, different from the case of CCR1, where the R$^{3.50}$ and Y$^{7.53}$ sidechains form a G protein binding interface as seen in typical class A GPCRs upon activation (Fig. 4B). Instead in UL33, F245$^{6.36}$ contacts Y306$^{7.53}$ to maintain the receptor conformation. In CX₃CR1, R$^{3.50}$ is also oriented inward where it maintains interaction with D$^{3.49}$ analogous to inactive class A GPCRs, while also forming a distinct interaction with T$^{2.39}$ at TM2. These structural features indicate that the N-terminal peptide does not stabilise the canonical active conformation for human class A GPCRs but the active state unique to UL33, which is more similar to the inactive state of typical class A GPCRs.

## Gα$_s$ C-terminal helix dominates the UL33-G$_s$ interface

As a result of the conformational similarity at the intracellular pocket UL33's G protein binding mode is closer to that of the CX₃CL1-CX₃CR1-$G_i$ complex (PDB: 7XBX) than the CCL15-CCR1-$G_i$ complex (PDB: 7VL9) (Fig. 5A), where the C-terminal hook at the end of the Gα$_s$-α5 helical terminal projects into a cavity around the TM7-H8 turn. UL33 exclusively interacts with the Gα subunit of heterotrimeric G$_s$ protein and has a Gα contact area of 920 Å². This area is slightly smaller than those observed in CCL15-CCR1-$G_i$ and CX₃CL1-CX₃CR1-$G_i$ with 1040 and 1110 Å² interface areas, respectively. In contrast the inactive US27-$G_i$ complex has a 650 Å² interface area (PDB: 7RKX, canonical-like state), reflecting incomplete insertion of Gα$_i$-α5 into the receptor core.

At the core UL33:Gα$_s$-α5 interface the L(−1) sidechain is recognised by the hydrophobic patch formed by M72$^{2.43}$, F245$^{6.36}$, and F313$^{8.50}$, while the L(−1) carboxyl tail interacts with G310$^{8.47}$ backbone amide (Fig. 5B left). L(−2) and E(−3) make sidechain-sidechain contacts with L63$^{ICL1}$ and T244$^{6.35}$, respectively. Y(−4) inserts between D128$^{3.49}$ and R129$^{3.50}$, forming a hydrogen bond and cation–π interaction, respectively. At the peripheral region ICL2 appears to contact Gα$_s$-αN, with R136$^{ICL2}$ likely contributing to

**Fig. 4 | Comparison of the activation mechanisms of UL33, CCR1, and CX₃CR1. A** Side and bottom views of the superimposed structure of UL33 (green), CCR1 (wheat, PDB: 7VL9), and CX₃CR1 (marine blue, PDB: 7XBX). The magenta double arrows indicate conformational differences at the intracellular ends of TM6 and TM7 of UL33/CX₃CR1 compared to CCR1 in active G protein-bound states and the purple double arrow shows the relative length at the TM6 bottom. **B** Comparison of the DR³·⁵⁰Y and NP⁷·⁵⁰xxY motifs among UL33, CCR1, and CX₃CR1 highlighting the 7TM conformation difference. The amino acid residues forming the motifs and interacting residues are shown as sticks and polar and other interactions are indicated by black and brown dashed lines, respectively.

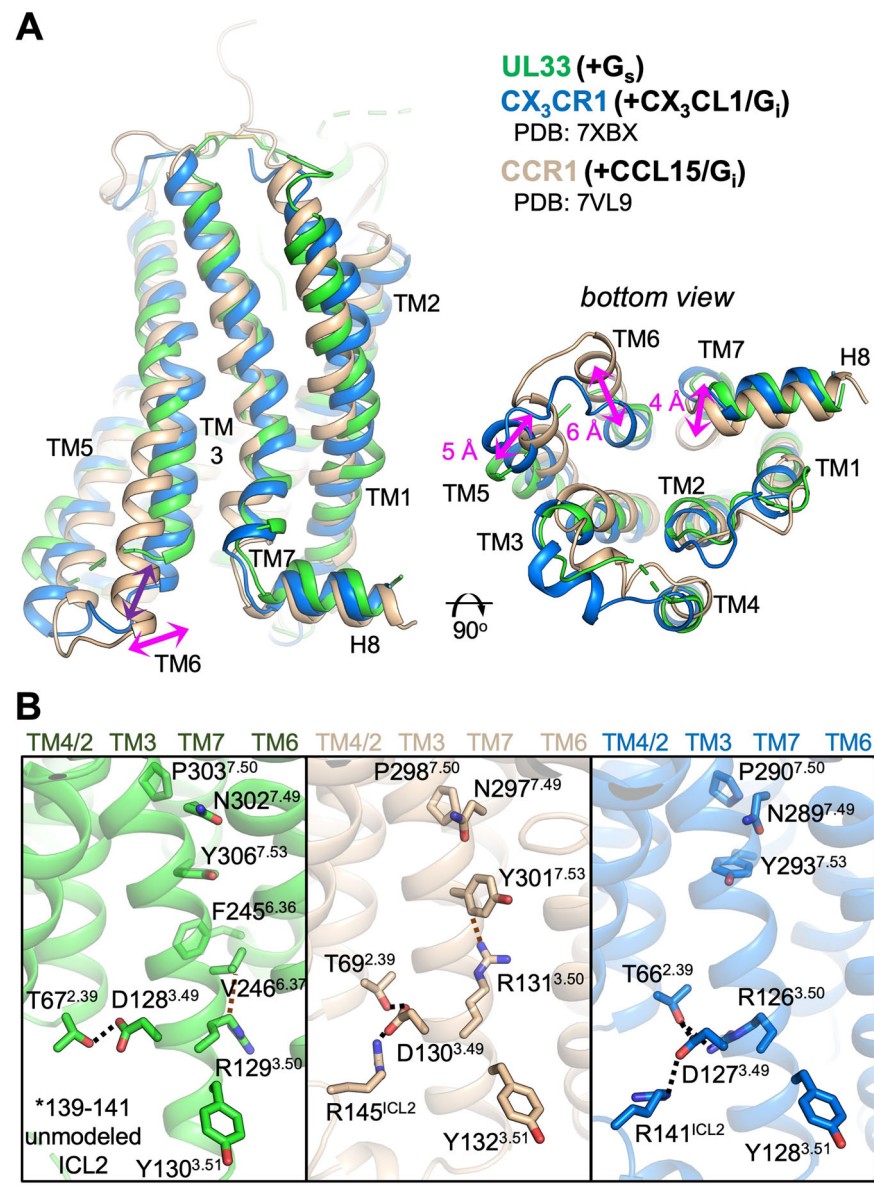

this interaction (Fig. 5B right). Furthermore the intracellular rim of TM3 and ICL3 appear to interact with the Gαs-α5 helix, where the α5 interface is formed by residues L(−7), Q384 and D381 of Gαs.

To map the hotspot residues for the direct UL33-Gs interaction we introduced single alanine mutations at the interface (Fig. 5C): D128A³·⁴⁹, R129A³·⁵⁰, V132A³·⁵³, R136A^ICL2, R233A^ICL3, and H311A^8·⁴⁸. Additionally the L133A³·⁵⁴ mutation was introduced outside the molecular interface but within the TM3-TM5 packing region (Fig. 5C). Note that R233 was not resolved in the structure model, but was hypothesised to contribute to the recognition of Gαs-α5 based on the blurred cryo-EM density. Notably disruption of the core interactions, D128³·⁴⁹ and R129³·⁵⁰ at the DRY motif, almost completely abolished UL33-mediated CREB signalling, and the V132³·⁵³ and H311⁸·⁴⁸ mutations resulted in large reductions in signalling but to a lesser extent than the DRY motif mutations. On the other hand R136A^ICL2 and R233A^ICL3 marginally decreased the Gs signalling, with only qualitative reduction for R136A^ICL2, essentially comparable to L133A located outside of the UL33:Gαs-α5 interface. Interestingly all mutants with impaired signalling showed elevated surface and total expression (Fig. 5C bottom; Supplementary Fig. 6C), although the magnitude of this increase did not correlate with the degree of functional loss. This suggests that, while reduced G protein coupling generally inhibits receptor downregulation, the

final expression level likely reflects a complex interplay between the extent of functional uncoupling and other mutation-specific effects on protein stability or trafficking.

These results indicate that core α5 interaction is a critical determinant for UL33-Gs signalling while the edge interactions between the ICLs and the globular domain of Gαs are auxiliary. This stands in contrast to the human chemokine receptor-Gi complexes where the α-helical ICL2 plays a major role (Fig. 5A)[50,51]. The less extensive contributions of the ICLs likely account for the observed structural flexibility between UL33 and Gs. Given that the structural flexibility was also observed or even more pronounced in complexes with Gq and Gi, the shallow G protein insertion might be a common feature of the three UL33-G protein complexes.

## Structural basis for G protein coupling profile of UL33

The interactions at the Gα C-terminus provide a structural rationale for UL33's G protein coupling profile (Gs/Gq/Gi). The sequence of the C-terminal five residues, especially at the −1 and −4 positions, of Gαs (Q**Y**EL**L**) shares relatively high similarity with Gαq (E**Y**NL**V**) but exhibits lower similarity to Gαi (D**C**GL**F**) and Gα₁₂/₁₃ (D**I**ML**Q** or Q**L**ML**Q**) (Fig. 6A). The binding pocket in UL33 appears optimised for the C-terminal

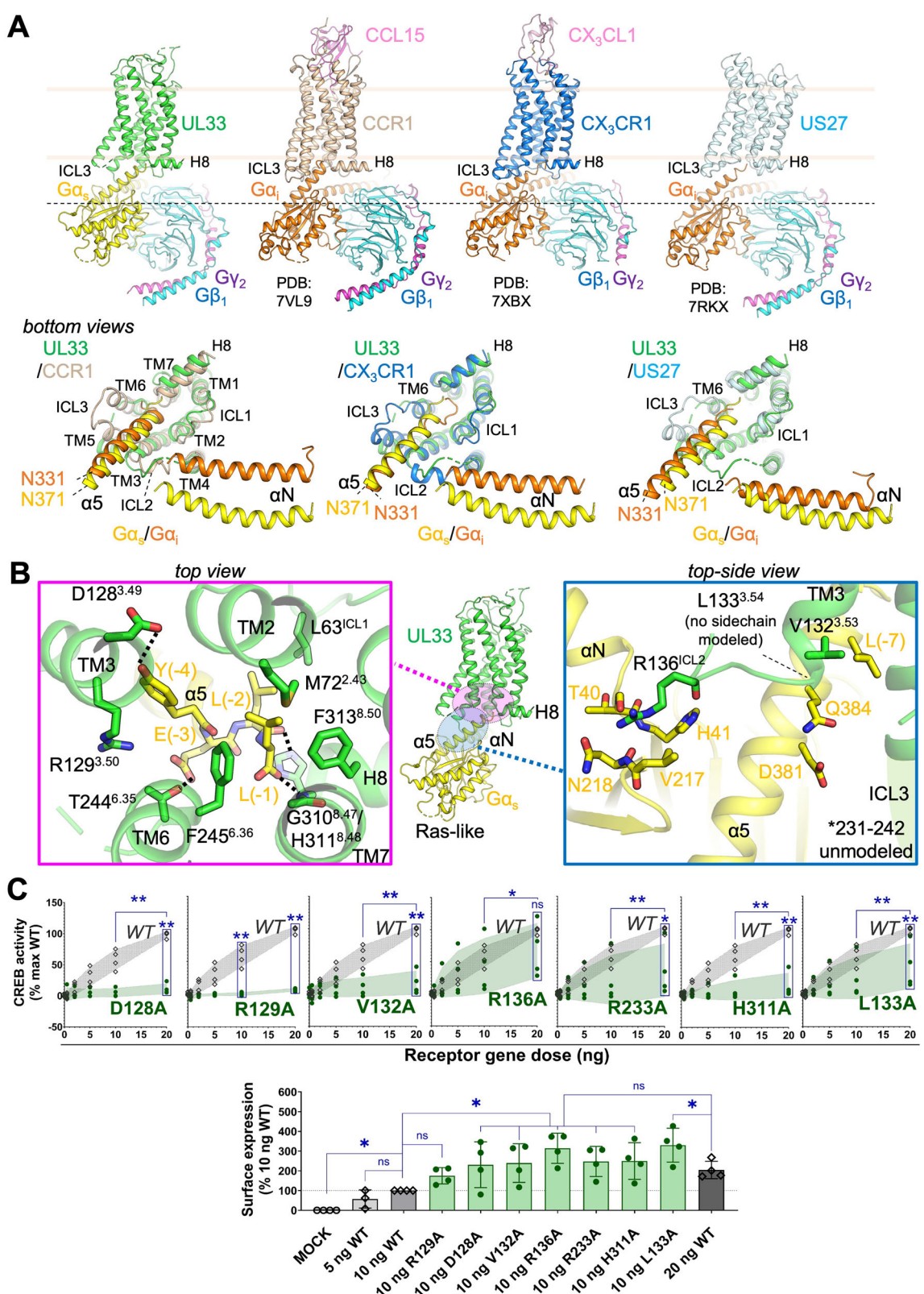

features shared by Gαs and Gαq. For example the aliphatic residue at the −1 position [L(−1)] of Gαs is recognised by a hydrophobic pocket formed by M72$^{2.43}$, F245$^{6.36}$, and F313$^{8.50}$ (Fig. 5B). This pocket appears flexible or spacious enough to accommodate the valine [V(−1)] of Gαq or the phenylalanine [F(−1)] of Gαi, but may be less favourable for the polar glutamine [Q(−1)] of Gα$_{12/13}$ (Fig. 6A, B). Additionally the −4 position of Gαs

[Y(−4)], which is shared with Gαq, forms extensive contact at the critical DR$^{3.50}$Y motif of UL33 (Fig. 5B, C).

Furthermore this mode of G protein engagement is distinct from that of other promiscuous viral GPCRs (Fig. 6B). For instance HCMV-US28, which also couples to Gq and Gi and G$_{12/13}$, but not to Gs, utilises a widely open intracellular core to engage Gαi more deeply[21]. Similarly Kaposi's

**Fig. 5 | Comparison of the G protein binding modes among UL33, CCR1, CX₃CR1, and US27 and analysis of the UL33-G_s interface. A** The top panel shows a side view of UL33-G_s, CCL15-CCR1-G_i (PDB: 7VL9), CX₃CL1-CX₃CR1-G_i (PDB: 7XBX), and US27-G_i (PDB: 7RKX). Nb35 or scFv16 included in the structures, are omitted for clarity. The bottom panel shows the superimposition of UL33-G_s with the three different GPCR-G protein complexes as indicated. Structural alignments were performed using Cα atoms of the receptors and only αN and α5 helices of Gα proteins are shown for clarity. **B** The amino acid residues mediating the major UL33:Gα_s interface. The left window shows the central UL33-Gα_s-α5 interface while the right panel shows the peripheral interacting region. The overall UL33-Gα_s structure was shown in the centre to indicate the approximate region magnified on both the left and right sides. The major interface residues are shown as sticks and polar and other interactions are indicated by black dashed lines, respectively. **C** The consequences of alanine mutations of UL33 at or near the Gα_s interface for its CREB activity and cell surface expression in HEK293A cells. CREB signalling activity was verified for the following UL33 mutants (represented in green circles in the individual plots): D128A, R129A, V132A, L133A, R136A, R233A, and H311A, and compared with the wild type activity (WT, grey open squares), across receptor gene doses ranging from 0 to 20 ng of DNA per 35,000 cells. The signalling results were obtained from $n = 4$ biologically independent experiments each performed in technical triplicate. Data were normalised to mock transfection (0% signalling) and the highest gene dose of WT UL33 (100% signal). Results are presented as individual data points for each experiment superimposed on a shaded area depicting the standard deviation (grey for WT and light green for mutants). The surface receptor expression data were also obtained from $n = 4$ biologically independent experiments each performed in technical triplicate, using a 10 ng gene dose for the WT and mutants per 35,000 cells. Expression results are presented individually normalised to mock transfection (0%) and the corresponding WT expression (100%). Data represent mean ± standard deviations. The differences in CREB activity between each of the UL33 mutants and WT at the highest gene dose as well as between gene doses ensuring similar surface expression levels of WT UL33 and its mutants, were determined using two-way ANOVA with Fisher's LSD post hoc test. Significant differences in expression levels between gene doses of WT UL33 and its mutants were determined using one-way ANOVA and Fisher's LSD post hoc test. Data sets compared between WT and a mutant are marked with an open rectangle when originating from the same gene dose and with a bracket when the gene doses of WT and the mutant are different. In all cases statistical significance is indicated as follows: *$P < 0.05$, ** $P < 0.01$, ns–not significant.

sarcoma-associated herpesvirus (KSHV)-encoded ORF74 adopts a similar active-state conformation to achieve a US28-like coupling profile[52]. In these G_s-excluding structures G(−3) faces an electronegative cage formed by carbonyls of TM7, which rejects Gα_s by repelling its acidic glutamate at the −3 position [E(−3)]. Interestingly in the UL33-G_s structure, this clash is avoided; a shallower α5 insertion orients E(−3) outward from the core region, thereby bypassing potential electrostatic repulsion with the electronegative cage. By contrast EBV-BILF1 recognises Gα_i by a mechanism different from that of US28 and ORF74, such that Gα_i-G(−3) directly contacts a methionine residue at TM2, which sterically rejects other Gα subunits with bulkier sidechains at the −3 position[31]. This comparison reveals that viral GPCRs have evolved diverse structural solutions to achieve specific G protein coupling profiles.

## Discussion

The structure of the UL33-G_s complex reveals a unique mechanism of constitutive activation coupled with a non-canonical mode of G protein engagement. The receptor's nascent N-terminal peptide functions as a tethered self-agonist, a mechanism that simultaneously occludes multiple chemokine-binding sites and explains UL33's orphan status. On the intracellular side UL33 displays an inwardly retained TM6, while still facilitating promiscuous coupling, thereby broadening its control over the host cell depending on the endogenous G protein expression patterns. This innate activation machinery is a key evolutionary divergence from typical chemokine receptors[6].

The atypical active conformation of UL33 closely resembles that of the active human CX₃CR1 (Fig. 4A) and remains globally similar to the inactive state of typical class A GPCRs[6]. Generally full GPCR activation by agonists involves a significant outward movement of TM6, which can range from moderate to large, opening the intracellular pocket[53]. This conformational change facilitates efficient G protein engagement and may also promote promiscuity by increasing the spatial accessibility for diverse G proteins. In sharp contrast the compact structure of UL33 may restrict the interaction interface with G proteins, potentially reducing signalling efficacy on a per-receptor basis[2]. Despite the narrow intracellular cavity with inwardly retained TM6, UL33 has acquired the ability to promiscuously activate multiple G proteins[3].

This seemingly paradoxical feature is explained by a precise recognition mechanism at the core interface allowing UL33 to stably engage G_s and G_q, tolerate G_i, yet strictly exclude G_{12/13}. This selectivity, driven by specific hydrophobic and electrostatic contacts at the 'wavy hook' of Gα C-terminus, provides a structural basis for the observed signalling profile of UL33, where it potently activates G_s and G_q (pathways with well-characterised functional roles[4,5]), activates G_i more weakly, and excludes G_{12/13}. This mechanism starkly contrasts with those of other promiscuous viral GPCRs like HCMV-US28 and KSHV-ORF74 with widely open TM6 (Fig. 6B). These findings underscore that viral GPCRs have evolved diverse structural solutions to achieve G protein promiscuity and suggest that UL33 has evolved to prioritise coupling to this specific subset of G proteins (G_s, G_i and G_q) for viral pathogenesis, even at the expense of maximal signalling efficiency.

From HCMV's strategic perspective this 'moderate' activation mode may be advantageous. Excessive disruption of host signalling pathways could induce cellular stress responses which might be detrimental to persistent viral infection. By maintaining a lower overall signal intensity—analogous to equipping CREB signalling with a moderate accelerator (G_s-coupling) and a weaker brake (G_i-coupling)—the virus can sustainably 'fine-tune' the host cellular environment. Since UL33 is constitutively active the virus can compensate for its lower intrinsic efficacy by regulating receptor expression levels to ensure sufficient overall signal magnitude. Indeed, loss-of-function mutants tend to show increased expression levels (Figs. 2D and 5C).

Furthermore this G protein coupling promiscuity is crucial for UL33 to adapt to changes in the host cellular environment. For instance during the viral lifecycle, when the cellular G protein balance shifts from a G_i-biased state towards a G_s-biased state, UL33 is poised to exploit this change. By leveraging this shift UL33 can immediately enhance G_s-mediated CREB activation and consequently act as an efficient trigger for viral reactivation.

While the tethered agonist mechanism presented here is conceptually similar to that of host PARs and aGPCRs[54,55], UL33 introduces a critical distinction. Its agonist is hard-wired from protein synthesis dispensing with the regulatory proteolytic unmasking required by PARs and aGPCRs. This ensures robust manipulation of host signalling pathways critical for the viral lifecycle such as G_s-mediated reactivation and G_q-dependent lytic replication[4,5]. The strategy is one of several convergent evolutionary paths herpesviruses have taken to acquire constitutive GPCRs[6,52]. Other vGPCRs employ similar but distinct approaches to utilise an extracellular loop agonist (EBV-BILF1 and KSHV-ORF74)[31,52,56] with variations in their reliance on transmembrane core mutations, highlighting a diverse toolkit of pathogenic modulators.

Finally our structural findings highlight potential avenues for therapeutic intervention. We identified a distinct druggable site at the interface between the tethered agonist and the transmembrane domain accessible via a narrow tunnel. This architecture suggests that molecules entering through the tunnel could engage a composite site affecting both the empty pocket and the adjacent N-terminal peptide. Targeting this unique viral interface presents a strategy to modulate UL33 activity with a potentially reduced risk of off-target effects. However the pharmacological outcome remains uncertain; depending on the precise interactions, such molecules might either disrupt the active conformation (negative modulation) or inadvertently stabilise it (positive modulation). Furthermore the observed

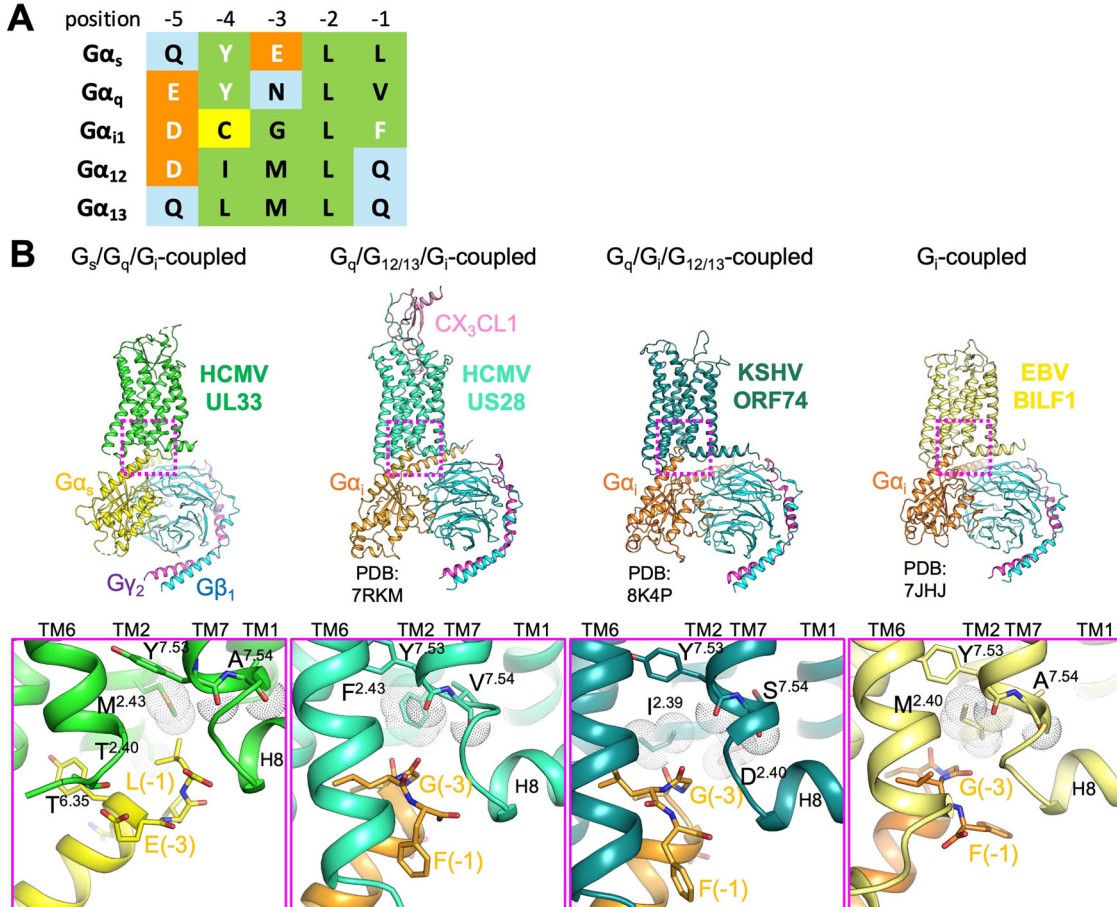

**Fig. 6 | Comparison of the G protein binding modes among signalling viral GPCRs. A** Sequence alignment of the C-terminal five residues of human Gα subunits. Residues are coloured by chemical properties (acidic: orange, hydrophobic or aromatic: green, polar: light blue, cysteine: yellow). **B** Comparison of the core G protein coupling interfaces in four herpesvirus GPCRs. The top panels show the overall structures: HCMV UL33-human $G_s$ (this study, PDB: 9WEY), human CX$_3$CL1-bound HCMV US28-human $G_i$ (PDB: 7RKM), KSHV ORF74-human $G_i$ (PDB: 8K4P), and EBV BILF1-human $G_i$ (PDB: 7JHJ). The G protein selectivity profile for each receptor is indicated above the structure with the G protein subtypes listed in descending order of their proposed primary coupling preference. The bottom panels show close-up views of the interaction interfaces with the C-terminal α-helix of the Gα subunits. The five Gα C-terminal residues and key residues in the receptor core (TM2, TM7, H8) are shown as sticks. The sidechains around the TM2-TM7-H8 pocket are also represented as dots to indicate their bulkiness.

structural flexibility and relatively weak density of the N-terminus suggest a dynamic interaction with the binding pocket. Leveraging this dynamic nature, strategies aimed at directly displacing the peptide might offer an alternative inhibitory approach. While the path toward therapeutic application requires careful validation, the detailed mechanistic insights presented here significantly advance our understanding of the HCMV 'GPCRome' and provide a structural foundation for future studies on viral GPCR function and inhibition.

## Methods
### Preparation of Nb35
Nb35[28] was cloned into pET26b(+) vector (Novagen) with C-terminal tandem 6×His tags. Nb35 was expressed in *E. coli* BL21(DE3) and purified from cell lysate with cobalt chelate chromatography using TALON resin (Takara Bio). The elution in HEPES-buffered saline (HBS, 10 mM HEPES-Na pH 7.4, 150 mM NaCl) containing 250 mM imidazole was concentrated to approximately 400 μM, supplemented with 10% (v/v) glycerol, flash frozen with liquid nitrogen, and stored at −80 °C for future use.

### Preparation of the UL33-G$_s$-Nb35 complex
The coding sequence of the full-length UL33 (UniProt ID: P16849) was cloned into pcDNA5/TO vector (Thermo Fisher Scientific) with the C-terminal Protein C epitope and LgBiT split NanoLuc fragment[27]. Gα$_s$

subunit with dominant negative mutations (S54N, G226A, E268A, N271K, K274D, R280K, T284D, I285T, A366S)[57], Gβ$_1$ with C-terminal HiBiT NanoLuc complementation fragment and N-terminal 3C protease-cleavable 10×His tag, and Gγ$_2$ were each cloned into the pEG vectors[58] for transient transfection (Gβ$_1$ and Gγ$_2$ vectors were a gift from Yuki Shiimura at Kurume University).

Expi293F inducible (Expi293Fi) cells (Thermo Fisher Scientific) were maintained in HE400AZ medium (GMEP) at a density between $0.3 \times 10^6$ and $4 \times 10^6$/mL and shaken at 37 °C and 130 rpm under 8% (v/v) CO$_2$. Induced and constitutive expression of UL33 and heterotrimeric G protein respectively, were performed similarly to our recent report[59] with polyethylenimine (PEI) MAX (Polysciences) as a transfection reagent[60], 3 mM sodium valproate and 0.4% (w/v) glucose as enhancers[61], and 8 μM doxycycline (Wako) as an inducer. The four plasmids were used at an equal weight ratio and a total of 1 μg of plasmid and 4 μg PEI per 1 mL cell culture were mixed in HE400AZ medum and incubated before transfecting the Expi293Fi cells. One day post-transfection, the enhancers were added and after an additional 24 h at 37 °C, the UL33 expression was induced by adding the inducer. The culture flask was transferred to a 30 °C incubator and shaken for 48 h. The cells were harvested, washed once with phosphate-buffered saline (PBS) and stored at −30 °C.

The whole cell pellet was directly solubilised in a buffer containing HBS, 20% (v/v) glycerol, 1% (w/v) n-Dodecyl-β-D-Maltoside (Anatrace),

0.1% (w/v) Lauryl Maltose Neopentyl Glycol (LMNG, Anatrace), 0.11% (w/v) Cholesteryl Hemisuccinate Tris Salt (CHS, Anatrace), cOmplete protease inhibitor cocktail (Roche), 10 mM $MgCl_2$, and 0.5 μL (0.25 unit) apyrase (New England Biolabs) for 2 h at 4 °C. After centrifugation at $15,000 \times g$ for 1 h at 4 °C the supernatant was supplemented with 3 mM $CaCl_2$ and in-house anti-Protein C Sepharose and incubated for 2 h at 4 °C. The resin was collected in a column and washed with the solubilisation buffer followed by HBS containing 0.1% LMNG/0.01% CHS and 3 mM $CaCl_2$, then HBS containing 0.01% LMNG/0.001% CHS and 3 mM $CaCl_2$. The complex was eluted from the column with HBS containing 0.001% LMNG/0.001% glyco-diosgenin (GDN, Anatrace)/0.0001% CHS, 5 mM EDTA, and 10 μg/mL Protein C peptide (PH Japan). The elution was concentrated to ~400 μL, quantified, and in-house 3C protease was added at a 1:50 (w/w) ratio. After overnight incubation at 4 °C the protein was purified by SEC using Superdex 200 Increase 10/300 GL (Cytiva) and the peak fractions were collected and mixed with 1.5 molar equivalents of Nb35 and the mixture was subjected to another SEC. The peak fractions were pooled and concentrated to approximately 9 mg/mL for cryo-EM specimen preparation.

### Cryo-EM grid preparation and data collection
The UL33-$G_s$-Nb35 complex (3 μL) was applied onto a glow-discharged 200 mesh copper R1.2/1.3 grid (Quantifoil) which was observed to possess 2 μm-diameter holes with 0.5 μm spacing. Grids were plunge-frozen in liquid ethane using a Vitrobot Mark IV (Thermo Fisher Scientific) with a 10-s hold period, blot force of 10 and blotting time of 3 s with 100% humidity at 4 °C. Cryo-EM data were collected using a JEM-Z320FHC electron microscope (JEOL) operated at 300 kV equipped with an omega-filter (20 eV slit width) and a K3 camera (Gatan) in the correlated double sampling mode. The calibrated pixel size was 0.79 Å at the specimen. Each movie was recorded for 2.6 s over 50 frames yielding a total exposure of 50 electrons/Å$^2$ and an exposure rate of 1 electron/Å$^2$ per frame. Two datasets were collected using SerialEM[62] with a nominal defocus range between −0.6 and −1.2 μm and with a 5 × 5 image shift pattern.

### Cryo-EM data processing
The data processing was carried out mainly in cryoSPARC[63] unless noted and the scheme is summarised in Supplementary Fig. 2. 6925 movies were collected as the first dataset and motion correction was performed with MotionCor2 (ref. [64]) wrapped in Relion4 (ref. [65]). Patch contrast transfer function (CTF) parameters were estimated for the motion-corrected micrographs and 4547 micrographs were selected based on the CTF values. Iterative particle picking (Gaussian picking, template-based picking and Topaz picking[66]), 2D classification, ab initio 3D reconstruction, and heterogeneous 3D reconstruction yielded 370,867 particles subjected to local non-uniform (NU) refinement. 3D classification was performed for this particle set without alignment revealing the conformational flexibility between the 7TM and G protein regions. Particles representing rarer views were retrieved from the less-defined 3D classes through 2D classification merged with the best 3D class, followed by a local NU refinement. Per-particle motion and dose weight were optimised using Relion's Bayesian polishing. The additional 2D classification and local NU refinement with CTF refinements led to a ~3.2 Å map with 223,543 polished particles (0.99 Å/pixel). Still the quality of the 7TM region was not readily interpretable for modelling, even with 3D flexible refinement or after another round of 3D classification, due to the variable local resolution. A total of 14,901 movies were then collected as the second dataset which was assessed on the fly by cryoSPARC Live, yielding 9268 movies for further analysis. The movies were motion-corrected with MotionCor2, and a similar data processing strategy was applied to yield 442,146 Bayesian-polished particles (0.99 Å/pixel). The polished particles were merged and subjected to 2D and 3D classifications similar to those described earlier giving the final 54,507 particles for the reconstruction of a 3.3 Å nominal resolution map with acceptable 7TM density. Homogeneous NU refinement appeared to yield a more interpretable map compared to locally NU-refined maps or 3D flexible refinement maps. Local NU-refinement and CTF refinements were

performed with the mask covering UL33. Local resolution was estimated using cryoSPARC.

### Model building
The ColabFold[46,67,68]-predicted UL33 structure and the Gs-Nb35 structure from the cryo-EM structure of serotonin 4 receptor-Gs-Nb35 (PDB ID: 7XT8) complex were used as starting models. Note that AlphaFold3 (ref. [36]) could not predict the N-terminal insertion until mid-2024. The model was manually fitted and corrected using Coot[69], idealised using ISOLDE[70] in UCSF ChimeraX[71] and refined in real space in the globally refined, uniformly sharpened map (−30 Å$^2$) using Phenix[72]. The model refinement process was performed iteratively until the model-map agreement was satisfied by visual inspection and model statistics had converged. The UL33-locally refined map was used as a modelling guide during visual inspection. The structure and map were visualised using UCSF ChimeraX and PyMOL (Schrödinger) where the secondary structure elements were assigned using the DSSP plugin[73]. The interface areas are estimated using PDBePISA[74].

### ColabFold prediction of UL33L
The full-length UL33L sequence was used for the model predictions by ColabFold[46] with the UL33 cryo-EM structure as a template. The five predicted structures were optimised using Amber[75] and RMSDs were calculated between each prediction and the experimental structure to determine the model for analysing the N-terminal protrusion. All other settings were set to default. The top-ranked prediction did not have the best pLDDT around the canonical N-terminal region and overall best RMSDs, and the third-ranked prediction was chosen. The structure was visualised using UCSF ChimeraX and secondary structures were assigned using the DSSP command.

### UL33 constructs for cell-based assay
The BBS-tagged[43] UL33 was designed based on the experimental structure and amino acid sequence where the region appears flexible and isolated from other areas of the receptor. With the BBS-tag insertion between T179 and N180 the predicted N-glycosylation sites remain unchanged. To facilitate correlation of receptor activity and expression studies an additional C-terminal Myc tag was also introduced. UL33 constructs UL33 WT, UL33-Myc, BBS-UL33, BBS-UL33-Myc and BBS-UL33 ΔD2-Myc were cloned into pcDNA3.1(+) by GenScript (Rijswijk, Netherlands). BBS-UL33-Myc alanine point mutations D2A, Y111A, D128A, R129A, V132A, L133A, R136A, R233A, R292A and H311A were generated using the polymerase chain reaction (PCR) overlap technique. The mutation was placed in the centre of the homologous primer pair flanked by 15 nucleotides on each side, ending with one or multiple cytosines/guanines and a GC content of 50–70%. The primer information is provided in the Supplementary Data. PCRs were performed with Phusion™ High-Fidelity PCR Master Mix with HF Buffer (NEB) and BBS-UL33-Myc in the pcDNA3.1(+) vector as template sequence according to the manufacturer's protocol, followed by DpnI (NEB) digest for 3 h at 37 °C. The mutations were verified by DNA sequencing (Eurofins Genomics).

### Mammalian cell culture for functional analysis
Human embryonic kidney (HEK) 293A cells (Thermo Fisher Scientific, Cat#: R70507, RRID: CVCL_6910) were cultured in high-glucose Dulbecco's Modified Eagle Medium (DMEM) supplemented with 10% (v/v) of heat-inactivated (56 °C for 30 min) fetal bovine serum and 1% penicillin-streptomycin and maintained at 37 °C in a 5% $CO_2$ humidified incubator. During seeding or passage the cells were detached using trypsin-EDTA.

### cAMP response element binding protein (CREB) activity assay
The cells were used to verify $G_s$-mediated signalling of UL33 and its mutants by measuring CREB activity as described previously[31,76]. Briefly HEK293A cells were seeded on poly-D-lysine pre-coated white 96-well plates (35,000 cells per well) and transfected with Lipofectamine 2000 in Opti-MEM reduced serum medium, according to the manufacturer's recommendations (Thermo Fisher Scientific). The cells were transfected with a range of

receptor gene doses (0–20 ng per well). Differences in receptor DNA amounts were compensated with pcDNA3.1(+) (empty vector). For cell signalling assessment the cells were co-transfected with the trans-reporting CREB system comprised of pFR-Luc trans-reporter plasmid (50 ng per well) and pFA2-CREB trans-activator plasmid (6 ng per well). After a 5-h transfection cells were allowed to regenerate in culture medium. Twenty-four hours after initiation of the transfection the cells were washed with PBS containing calcium and magnesium ions (PBS$^{++}$) and incubated with 100 μL of 2-fold dilution of SteadyLite Plus (PerkinElmer) in PBS$^{++}$ for 30 min, followed by luminescence emission measurement using EnVision Multilabel microplate reader (PerkinElmer). Unless stated otherwise the data were collected from four independent experiments, each performed in technical triplicate, normalised to mock-transfected cells (containing 0 ng of receptor DNA and referred to as 0% activity) and the highest dose of BBS-UL33-Myc receptor tested (20 ng per well, referred to as 100% WT activity) and presented individually for each experiment.

### Detection of UL33 receptor expression variants via fluorescence-activated cell sorting (FACS)

Fluorescence-activated cell sorting (FACS) was used to measure surface and total expression levels of UL33 variants in HEK293A cells, corresponding to CREB activation. Twenty-four hours before transfection 3.5 million cells were seeded per T75 flask. On the following day due to 100-fold larger sample size compared to the transfections for the CREB activity assay, the cells were transfected with PEI reagent and proportionally higher amounts of DNA: 0–2 μg of receptor DNA with corresponding vector DNA supplementation, 5 μg of pFR-LUC and 0.6 μg of pFA2-CREB (total amount of DNA summed up to 7.6 μg per 3.5 million cells). One microgram of receptor DNA per sample was used for expression comparison between WT and UL33 mutants. The DNA:PEI ratio of 1:2 was applied (corresponding to 15.2 μg of PEI per 3.5 million cells)[77]. The PEI was preincubated in 5 mL of regular culture medium for 5 min at room temperature prior to the addition of the DNA mix. The DNA-PEI mixture in culture medium was incubated for an additional 15 min. In the meantime 5 mL of fresh culture medium was added to the cells per T75 flask. Afterwards the DNA-PEI mix in culture medium was added dropwise to the cells and incubated for 24 h. The next day the transfection mixes were removed from the cells, substituted with 5 mL of growth medium per flask, and incubated for another 24 h before harvesting.

On the following day the cells were washed with calcium- and magnesium-free Hank's balanced salt solution (HBSS$^-$, Gibco), scraped, resuspended in HBSS$^-$ to a final density of 1 million cells per mL and portioned into 1 mL samples. The cells were further stained with 0.2 μL LIVE/DEAD™ Fixable Blue Dead Cell Stain (Cat. No. L23105, Thermo Fisher Scientific) per 1 million cells for 30 min at room temperature in the dark. Following viability staining cells were washed once with HBSS$^-$ and fixed with 500 μL of 4% paraformaldehyde in HBSS$^-$ per sample for 10 min. Fixed cells were washed three times with HBSS$^-$ and centrifuged at 1200 rpm for 2 min between washes. For total receptor detection cells were permeabilised with 200 μL of 0.1% Triton X-100 per 1 million cells for 20 min at room temperature, followed by three additional washes. Cells were then blocked with 1 mL of HBSS$^-$ containing 1% bovine serum albumin (BSA) for 30 min at room temperature. Staining was performed by incubating cells with 0.5 μL Alexa Fluor 488-conjugated α-bungarotoxin (BTX-AF488; Cat. No. B13422, Thermo Fisher Scientific) per 1 million cells for 2 h at 4 °C in the dark. After staining cells were washed three times and resuspended in 200 μL HBSS$^-$ containing 1% BSA per sample for flow cytometric analysis.

Expression data acquisition was performed on a FACSDiscover™ S8 Cell Sorter (BD Biosciences) and analysis was carried out using FCS Express 7 Research Edition (De Novo Software). The gating strategy used for viability and receptor expression analysis is detailed in Supplementary Fig. 7. Data were collected from 3 to 4 independent experiments from at least 5000 cells per sample, registered as median fluorescence intensity (MFI), normalised to signal obtained from mock-transfected cells (0% receptor

expression) and cells transfected with 1 μg of WT UL33 per sample (100% of WT expression) and presented as individual values or as means correlated with CREB activity data. Representative pictures of cells expressing AF488-BTX-stained receptor are presented in Supplementary Fig. 6.

### Statistics and reproducibility

All functional data regarding receptor signalling and expression were obtained from at least three biologically independent experiments ($n \geq 3$). The data were consistent across experiments and statistical analyses were performed to determine the significance of the results.

Statistical analyses were performed using GraphPad Prism (GraphPad Software, version 10). One-way ANOVA followed by Fisher's least significant difference post hoc test was applied to determine differences between the activities of WT UL33 receptor gene doses and mock transfection, as well as between surface expression levels of WT UL33 gene doses and UL33 alanine mutants. Two-way ANOVA was applied to assess significant differences between pairs of CREB signals obtained for the highest receptor gene doses of WT and mutant UL33 and signals obtained from the WT and mutant UL33 expressed on the cell surface in similar amounts. In all cases significant differences are indicated as *$P < 0.05$ and **$P < 0.01$. Non-significant differences are noted as 'ns'.

### Reporting summary

Further information on research design is available in the Nature Portfolio Reporting Summary linked to this article.

### Data availability

The cryo-EM maps and model coordinate file of the UL33-G$_s$-Nb35 complex have been deposited in the Electron Microscopy Data Bank (EMDB) under accession code EMD-65918, and in the Protein Data Bank (PDB) under accession code 9WEY, respectively. The pcDNA3.1(+) BBS-UL33-Myc plasmid has been deposited to Addgene (Plasmid #251747). All other data required to evaluate the conclusions are included in the manuscript. The Source data behind the figures in the paper are available in Supplementary Data. Any remaining information can be obtained from the corresponding author upon reasonable request.

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

## Acknowledgements

This work was partly supported by the Japan Society for the Promotion of Science (JSPS) KAKENHI (JP22K20632/JP24K01965/JP24K21935 to N.T., JP20H00451 to Y.F., JP24K18061 to S.S.), the Japan Agency for Medical Research and Development (AMED) under Grant Number JP21ae0121028 for Y.F., the Danish Council for Independent Research (6110-00688B to M.M.R.), the European Research Council (VIREX: 682549 and MEDICATE: 101055152 to M.M.R.), the Novo Nordisk Foundation (NNF20OC0062899 to M.M.R.), the Lundbeck Foundation (R242-2017-409 to M.M.R.), and the B-ACTIVE (project number 101120187) under the Horizon Europe pro-gramme (HORIZON-MSCA-2022-DN-01 to M.M.R. and C.W.). N.T. was supported by the Nakajima Foundation to carry out this study. Graphics presenting UL33 signalling and principles of related assays were created in BioRender.com. We acknowledge Gelo Victoriano Dela Cruz from the Novo Nordisk Foundation Centre for Stem Cell Medicine, reNEW, University of Copenhagen, Denmark, for excellent assistance in flow cytometry. We also thank the members of the Cellular and Structural Physiology Laboratory (CeSPL) for daily scientific discussions and Ms. Miho Sasaki for her research support.

## Author contributions

A.K.D.: data curation (lead), formal analysis (lead), investigation (lead), methodology (supporting), validation (equal), visualisation (supporting), writing–original draft (equal), writing–review and editing (equal). S.S.: investigation (supporting), software (equal), writing–review and editing (supporting). C.W.: formal analysis (supporting), investigation (supporting), visualisation (supporting), writing–review and editing (supporting). F.F., K.N. and A.K.: investigation (supporting), writing–review and editing (supporting). Y.F.: funding acquisition (lead), methodology (equal), resources (lead), supervision (supporting), writing–review and editing (supporting). M.M.R.: funding acquisition (equal), methodology (equal), project administration (equal), resources (lead), supervision (equal), writing–original draft (equal), writing–review and editing (equal). N.T.: conceptualisation (lead), data curation (lead), formal analysis (supporting), funding acquisition (equal), investigation (lead), methodology (equal), project administration (equal), resources (supporting), software (equal), supervision (equal), validation (equal), visualisation (lead), writing–original draft (lead), writing–review and editing (lead).

## Competing interests

The authors declare the following competing interests: M.M.R. is a co-founder and board member of Synklino A/S; however, the work presented here is independent of her role at Synklino A/S. C.W. performed the research during her 6-month secondment from Synklino A/S in the laboratory of M.M.R. at the University of Copenhagen. The other authors declare no competing interests.
