## [Transparent Peer Review File · Communications Biology]

Activation of cytomegalovirus-encoded G protein-coupled receptor UL33 by an innate N-terminal peptide

Corresponding Author: Dr Naotaka Tsutsumi

Version 0:

Reviewer comments:

Reviewer #1

(Remarks to the Author)

This manuscript by Drzazga et al. presents the cryo-EM structure of the UL33-Gs protein complex, with its N-terminal peptide functioning as a tethered agonist. The authors reveal that the tethered peptide adopting a β -sheet conformation forms a parallel three-stranded β -sheet together with ECL2 and extends the ligand-binding pocket of UL33. Interestingly, UL33 displays an inwardly-retained TM6, which contrasts with that of most other activated CCRs and typical class A GPCRs. Together, these findings expand our knowledge about the unique activation mechanism for UL33, especially the mechanism for the viral GPCR to adapt to the host cellular environment. However, there are still some minor comments that need to be addressed.

1. Are there differences in detecting the coupling activity of UL33 to Gs, Gi, and Gq, given that only the UL33-Gs-Nb35 complex has been resolved?
2. In fig.2A, It would be beneficial to denote the location of the 'major' pocket more clearly in the cross-sectional view (like fig.3B).
3. It is better to add the TM number in figure5A bottom views.
4. Line 206: What are the respective outward movement distances of TM5 and TM7?

Reviewer #2

(Remarks to the Author)

The manuscript "Activation of cytomegalovirus-encoded G protein coupled receptor UL33 by an innate N-terminal peptide" by Drzazga, A.K. et al. describes the CryoEM of the UL33 viral GPCR in an active state with the N-terminal fragment acting as the orthosteric agonist. The authors provide both functional and structural data to support their claims. In general, I found the manuscript to be well written with few flaws regarding methodology and interpretation of their results. The authors provided their current pdb file and CryoEM maps for inspection. My recommendation is to accept the manuscript with minor revisions.

Below are my minor comments:

1. The cryoem maps are sufficient to interpret many of the findings the authors describe in their manuscript but some areas are a bit ambiguous. I would suggest to further refine the structure using local refinement and a mask around the receptor with a second local refinement around the g-protein, this should improve the quality of the map and make some areas more interpretable. Additionally, the use of b-factor sharpening with programs such as deepEMhancer would also help. Furthermore, there appears to be several cholesterol that could be modeled on the exterior of the 7TM.
2. The authors extensively describe through comparison different aspects the UL33 GPCR with that of chemokine and PAR GPCR structures. Little is discussed about viral GPCR comparisons, I believe a comparison with other viral GPCR structures would be warranted.
3. Page 6, line 201, could a reference for the comment be included.
4. Is this virus restricted to a specific cell type and if so would that affect the coupling to specific G proteins? Thus type of

discussion extends from the authors comments regarding G protein specificity.

Reviewer #3

(Remarks to the Author)

The manuscript Drzazga et al. features the cryo-EM SPA structure of the constitutively active viral receptor UL33, which may represent a target for new antiviral therapy. Density within the orthosteric region is consistent with its N-terminus, which appears to trap the receptor in an active state. Interestingly, this active receptor adopts an unusual cytoplasmic configuration with TM6 not extended outwards from the core as in canonical class A GPCRs. However, the docking of the G protein gives insight into why the receptor is promiscuous when it comes to the heterotrimeric G protein it would interact with. The authors go on to validate the interactions they see using site directed mutagenesis and CREB-based transcriptional assays. Overall, the paper is well written, the figures are lovely, and the data overall looks to be sound. Repts and statistics are described well, and so the work seems to be rigorous.

Issues/considerations:

1) The density for the N-terminal peptide is of poor quality. It is good that the authors checked some of the contacts observed to help support their interpretation. The density shown for the N-terminal peptide in Fig S3D is misleading because at the angle shown it doesn't reveal that the backbone of the first four residues do not have continuous density; please generated a less biased view and move that to the main text. It is important that readers understand how tenuous the assignment of structure to this map region really is. I think it is more or less correct, but one doesn't get a sense of the dynamics the way the authors show the data. Likely there is a high degree of disorder in this region and the authors could consider being more circumspect in terms of describing what residues are interacting in the model. Overall the model and map show good correlation, but the authors should consider deleting the regions where there is no interpretable density, such as most of ICL3 and perhaps even ICL2. By showing complete models for poor density regions it implies that these areas can be built with high confidence, they cannot.

2) In the abstract and the discussion, the authors put forward the claim that there is enough room in the receptor N-terminus binding cleft that one might be able to identify negative allosteric modulators that might treat viral infections. This is a nice idea, but a highly speculative one and as such I don't think it is appropriate for the abstract. Fine in the discussion. Consider deleting this from the abstract. Moreover, because the receptor N-terminus is exerting high dynamics, the position of the atoms are not known well enough for one to use it for rational design.

Minor glitches:

Line 77. "narrow cavity to the empty" should be "narrow cavity next to the empty"?

Lines 96-98. "However, only the UL33-Gs-Nb35 complex yielded interpretable 3D reconstruction, indicating that the interaction between UL33 and the G protein is structurally flexible". I don't see why the the fact that this particular complex (as opposed to those for Gq, etc) could give interpretable cryo-EM density necessarily means that the interaction between the receptor and G protein is flexible.

line 238. "loose cation- π stacking". Readers are not going to know what "loose" means in this context, and is the density really good enough for the authors to say whether it is strong or weak. In general, avoid these kinds of fuzzy adjectives. Moreover, limit descriptions of these kinds of details, because I do not believe one can see these details in all regions of the maps.

lines 256-258. "These results indicate that core $\alpha 5$ interaction is a critical determinant for UL33-Gs signaling, and the edge interactions between ICLs and the globular domain of G α s are supportive, consistent with the observed structural flexibility between UL33 and Gs." Again, I don't see how it follows that these interactions are consistent with observed structural flexibility.

Lines 277-279." Indeed, our cryo-EM analysis suggested structural flexibility between UL33 and the G proteins (Fig. S1; Fig. S2). Despite this, UL33 has acquired the ability to promiscuously couple to Gs, Gi, and Gq3." Why "despite"? wouldn't more flexibility allow a receptor to accommodate a larger number of G proteins? I also do not see how Figures S1 and S2 necessarily indicate high flexibility.

Final paragraph. I do not really follow the logic. Molecules that would bind here may more likely be PAMs than NAMs. I would predict most molecules that bind in there would keep the N-terminal methionine in its pocket and thus keep on activating the receptor. Seems like one would be better off finding compounds that displace the N-terminus because based on the density, it is not bound very tightly. In the end, the "allosteric modulator" aspect of the paper is not a strong selling point to me and the authors may want to tone that down.

Question: Although it clearly worked, the use of a fusion to the Gbeta1 C-terminus seems like a risky move given that the C-terminal carboxylate of the the protein forms structural interactions with the protein core. Is there any chance this unusual nanobit linkage might perturb the overall configuration between the G protein and the receptor? Also, the authors might want to explain why they went with Nanobit in the first place. Is it because they could not isolate a complex without it (i.e., did they try)?

Version 1:

Reviewer comments:

Reviewer #1

(Remarks to the Author)

In their revised manuscript, the authors have comprehensively responded to all points raised. The revisions, which satisfactorily address the reviewers' comments, have been incorporated and have strengthened the work. It could be considered for publication.

Reviewer #2

(Remarks to the Author)

I would like to thank the authors for addressing my comments from the initial review of their manuscript. I am satisfied with their answers to not only my questions but also the other reviewers' questions and comments and would like to recommend the manuscript for acceptance.

Reviewer #3

(Remarks to the Author)

The authors have satisfactorily addressed my comments. By the way, I did write for the last comment that the nano bit linkage in question was to the C-terminus of Gbeta in my comment to the authors (not Galpha as the rebuttal says). However, given that This region is relatively distant from the receptor interface I don't think it is a major issue. It's a strange choice, but I guess it works.

Reviewer #1 (Remarks to the Author):

This manuscript by Drzazga et al. presents the cryo-EM structure of the UL33-Gs protein complex, with its N-terminal peptide functioning as a tethered agonist. The authors reveal that the tethered peptide adopting a β -sheet conformation forms a parallel three-stranded β -sheet together with ECL2 and extends the ligand-binding pocket of UL33. Interestingly, UL33 displays an inwardly-retained TM6, which contrasts with that of most other activated CCRs and typical class A GPCRs. Together, these findings expand our knowledge about the unique activation mechanism for UL33, especially the mechanism for the viral GPCR to adapt to the host cellular environment. However, there are still some minor comments that need to be addressed.

We thank the reviewer for the positive assessment of our work and for the constructive comments, which have helped us improve the clarity and presentation of the manuscript.

1. Are there differences in detecting the coupling activity of UL33 to Gs, Gi, and Gq, given that only the UL33-Gs-Nb35 complex has been resolved?

We have not quantitatively assessed UL33's coupling efficiency for the different G proteins. Measuring these parameters for a constitutively active receptor presents significant challenges compared to assessing ligand effects. Experimentally, however, we observed a generally higher yield and a more symmetrical size-exclusion chromatography (SEC) peak for the UL33-G_s complex compared to UL33-G_i or UL33-G_q with the co-expression strategy as shown in Fig. S1. While these observations might suggest a preference for G_s coupling, we refrained from including this discussion in the manuscript as the data remains speculative. We appreciate the opportunity to share these experimental observations and interpretation here.

2. In fig.2A, It would be beneficial to denote the location of the 'major' pocket more clearly in the cross-sectional view (like fig.3B).

We thank the reviewer for this suggestion. We have prepared a dedicated figure (new Figure S4B) illustrating the locations of both the major (main) and minor pockets in a cross-sectional view to improve clarity. Here we chose a top view rather than the side view to effectively show the entire pockets.

3. It is better to add the TM number in figure5A bottom views.

We have added the TM numbers to the bottom views in Figure 5A as requested. To avoid crowding, we only added full labels to the left panel.

4. Line 206: What are the respective outward movement distances of TM5 and TM7?

We only resolved the constitutively active, G_s-coupled state of UL33, thus our data do not show the inactive-to-active structural transition. Instead, we added the differences in respective TM positions between the conformationally similar receptors UL33 and CX₃CR1, and CCR1, and indicated the numbers in both Figure 4A and the main text.

The corresponding sentence now reads (line 221 of the revised manuscript):

"However, at the intracellular ends, UL33's TM5 and TM7 are positioned 5 Å and 4 Å outward, respectively, while TM6 is inwardly retained, located 6 Å closer to the central axis compared to CCR1 (Fig. 4A)."

Reviewer #2 (Remarks to the Author):

The manuscript "Activation of cytomegalovirus-encoded G protein coupled receptor UL33 by an innate N-terminal peptide" by Drzazga, A.K. et al. describes the CryoEM of the UL33 viral GPCR in an viral GPCR active state with the N-terminal fragment acting as the orthosteric agonist. The authors provide both functional and structural data to support their claims. In general, I found the manuscript to be well written with few flaws regarding methodology and interpretation of their results. The authors provided their current pdb file and CryoEM maps for inspection. My recommendation is to accept the manuscript with minor revisions.

We thank the reviewer for the constructive review and positive recommendation.

Below are my minor comments:

1. *The cryoem maps are sufficient to interpret many of the findings the authors describe in their manuscript but some areas are a bit ambiguous. I would suggest to further refine the structure using local refinement and a mask around the receptor with a second local refinement around the g-protein, this should improve the quality of the map and make some areas more interpretable. Additionally, the use of b-factor sharpening with programs such as deepEMhancer would also help. Furthermore, there appears to be several cholesterol that could be modeled on the exterior of the 7TM.*

We appreciate the reviewer's suggestions for improving the map quality. We thoroughly explored these methodologies during data processing, but unfortunately, they did not substantially improve the interpretability of the flexible regions of the UL33-G_s complex.

1) Local refinement: We acknowledge this methodology often improves local map quality, and in our hand, worked well for many GPCR-G protein complexes. However, while we performed extensive testing of local refinement focused on the UL33 region using various parameters and masks, we observed little to no improvement in the interpretability of the transmembrane domain. Therefore, while the locally refined map was used as a reference for chain tracing, only the consensus, globally-refined map was used for the final refinement.

2) Map sharpening: We tested tools such as deepEMhancer and local anisotropic sharpening by Phenix. While deepEMhancer improved the overall appearance of the map, both tools introduced artificial features in critical local regions. To address issues with over-sharpening and balance the visualization of higher-resolution features with map connectivity, we sharpened the map with an overall B-factor of -30 Å². The model has been updated conservatively and re-refined against this map.

3) Cholesterol modeling: While we agree that there are several cholesterol-like densities present, we prefer to refrain from modeling small molecules in ambiguous density, as accurate identification and placement are more challenging than for the polypeptide chain.

2. *The authors extensively describe through comparison different aspects the UL33 GPCR with that of chemokine and PAR GPCR structures. Little is discussed about viral GPCR comparisons, I believe a comparison with other viral GPCR structures would be warranted.*

We agree that a comparison with other viral GPCRs is warranted. We have added a figure (new Figure 6 in the final subsection in the Results section, starting on line 285 of the revised manuscript) comparing the G protein binding modes between UL33-G_s and the related viral GPCRs for the discussion of their G protein selectivity. As the results section became lengthier, we added subheadings for improved readability. We have also revised a discussion of the activation mechanisms of G proteins, starting on line 328 of the revised manuscript.

3. Page 6, line 201, could a reference for the comment be included.

We have revised the preceding sentences to be more explicit and included the relevant citations. The sentence now reads (line 211 of the revised manuscript):

"In contrast, CCR1, a putative ancestor of UL33, consists of a single coding exon⁴⁷. Furthermore, many herpesvirus genes are intronless even when the original host gene contains this segment, a result of capturing intron-free cDNA copies of host mRNAs⁴⁸. These facts suggest that the shared two-exon structure of the UL33 gene family likely evolved..."

4. *Is this virus restricted to a specific cell type and if so would that affect the coupling to specific G proteins? This type of discussion extends from the authors comments regarding G protein specificity.*

This is a very interesting point regarding the interplay between viral tropism and G protein coupling specificity. While we acknowledge the importance of this question, we were unable to verify a clear trend from currently available public information. We believe this warrants further investigation in future studies.

Reviewer #3 (Remarks to the Author):

The manuscript Drzazga et al. features the cryo-EM SPA structure of the constitutively active viral receptor UL33, which may represent a target for new antiviral therapy. Density within the orthosteric region is consistent with its N-terminus, which appears to trap the receptor in an active state. Interestingly, this active receptor adopts an unusual cytoplasmic configuration with TM6 not extended outwards from the core as in canonical class A GPCRs. However, the docking of the G protein gives insight into why the receptor is promiscuous when it comes to the heterotrimeric G protein it would interact with. The authors go on to validate the interactions they see using site directed mutagenesis and CREB-based transcriptional assays. Overall, the paper is well written, the figures are lovely, and the data overall looks to be sound. Repts and statistics are described well, and so the work seems to be rigorous.

We thank the reviewer for recognizing the rigor of our study. We have addressed the concerns regarding the structural interpretation and the scope of our claims, as detailed below.

Issues/considerations:

1) *The density for the N-terminal peptide is of poor quality. It is good that the authors checked some of the contacts observed to help support their interpretation. The density shown for the N-terminal peptide in Fig S3D is misleading because at the angle shown it doesn't reveal that the backbone of the first four residues do not have continuous density; please generate a less biased view and move that to the main text. It is important that readers understand how tenuous the assignment of structure to this map region really is. I think it is more or less correct, but one doesn't get a sense of the dynamics the way the authors show the data. Likely there is a high degree of disorder in this region and the authors could consider being more circumspect in terms of describing what residues are interacting in the model. Overall the model and map show good correlation, but the authors should consider deleting the regions where there is no interpretable density, such as most of ICL3 and perhaps even ICL2. By showing complete models for poor density regions it implies that these areas can be built with high confidence, they cannot.*

As suggested, we added the cryo-EM density corresponding to the N-terminal region of UL33 to the main figure from a different angle (Figure 1B). The angle in the Fig. S3D remains similar to the previous version for complementarity. We also revised the model to be more conservative regarding the intracellular loops, removing the entire ICL3 and a part of ICL2. Note that we changed the map sharpening B factor to -30 \AA^2 to slightly alleviate fragmentation of the map.

Concerning the N-terminal region, our original interpretation was to attribute the limited map quality to the global flexibility between UL33 and G_s . However, as the reviewer pointed out, it might also imply local flexibility. We have thus toned down the statement regarding the N-terminal sheet formation by inserting the following sentence in line 108 of the revised manuscript:

"It should be noted that the cryo-EM density for this N-terminal peptide is of relatively poor quality (Fig. 1B left inset; Fig. S3D), implying a high degree of local flexibility or dynamics. Nevertheless, ..."

We also appreciate the reviewer's critical assessment regarding the modeling of the intracellular loops. The cryo-EM density in these areas is indeed weak, suggesting higher flexibility or metastability. Modeling in such areas is challenging, particularly in balancing the visualization of weak density against the risk of interpreting noise as signal. In our initial submission, we attempted to carefully trace the most plausible path for the backbone and

several sidechains by generally relying on features visible at a relatively low but acceptable contour level (up to approximately 3σ in a sharpened map and 4σ in an unsharpened map), with reasonable chemical interaction networks in mind. Importantly, this initial modeling effort was instrumental in identifying potential key residues within these loop regions and designing the subsequent mutagenesis experiments. Nevertheless, we fully agree with the reviewer that representing such dynamic regions with a single static model is misleading, even with a figure showing a local map resolution. Following the reviewer's advice, we have removed the most flexible segments in the intracellular loops (specifically, residues 231–242 in ICL3 and residues 139–141 in ICL2) from the revised model as shown in Fig. 5B and Fig. 4B, respectively.

2) In the abstract and the discussion, the authors put forward the claim that there is enough room in the receptor N-terminus binding cleft that one might be able to identify negative allosteric modulators that might treat viral infections. This is a nice idea, but a highly speculative one and as such I don't think it is appropriate for the abstract. Fine in the discussion. Consider deleting this from the abstract. Moreover, because the receptor N-terminus is exerting high dynamics, the position of the atoms are not known well enough for one to use it for rational design.

We agree that the claim regarding the potential for identifying negative allosteric modulators is rather speculative and only appropriate in the Discussion. As suggested, we have removed the last sentence from the abstract. Furthermore, acknowledging the high dynamics of the N-terminus and the inherent challenges this presents for rational drug design, we have significantly toned down the corresponding discussion in the main text.

The final paragraph now reads (line 365 of the revised manuscript):

"Finally, our structural findings highlight potential avenues for therapeutic intervention. We identified a distinct druggable site at the interface between the tethered agonist and the transmembrane domain, accessible via a narrow tunnel. This architecture suggests that molecules entering through the tunnel could engage a composite site, affecting both the empty pocket and the adjacent N-terminal peptide. Targeting this unique viral interface presents a strategy to modulate UL33 activity with a potentially reduced risk of off-target effects. However, the pharmacological outcome remains uncertain; depending on the precise interactions, such molecules might either disrupt the active conformation (negative modulation) or inadvertently stabilize it (positive modulation). Furthermore, the observed structural flexibility and relatively weak density of the N-terminus suggest a dynamic interaction with the binding pocket. Leveraging this dynamic nature, strategies aimed at directly displacing the peptide might offer an alternative inhibitory approach. While the path toward therapeutic application requires careful validation, the detailed mechanistic insights presented here significantly advance our understanding of the HCMV "GPCRome" and provide a structural foundation for future studies on viral GPCR function and inhibition."

Minor glitches:

Line 77. "narrow cavity to the empty" should be "narrow cavity next to the empty"?

We have updated the text to "narrow cavity **connecting** to the empty pocket".

Lines 96-98. "However, only the UL33-Gs-Nb35 complex yielded interpretable 3D reconstruction, indicating that the interaction between UL33 and the G protein is structurally flexible". I don't see why the fact that this particular complex (as opposed to those for Gq, etc) could give interpretable cryo-EM density necessarily means that the interaction between the receptor and G protein is flexible.

We apologize for the confusing phrasing. We intended to convey that the inability to obtain interpretable reconstructions for G_i or G_q suggests that the interactions between UL33 and those G proteins are structurally flexible. We have revised the sentence to clarify this point. We also added the statement "suggesting that the UL33-G_s interface is also tenuous" to the following sentence, based on the variable local resolution observed in the map.

line 238. "loose cation- π stacking". Readers are not going to know what "loose" means in this context, and is the density really good enough for the authors to say whether it is strong or weak. In general, avoid these kinds of fuzzy adjectives. Moreover, limit descriptions of these kinds of details, because I do not believe one can see these details in all regions of the maps.

We apologize for the use of the fuzzy adjective "loose". We have revised this description to "cation- π interaction" and omitted several detailed explanations of interactions.

lines 256-258. "These results indicate that core $\alpha 5$ interaction is a critical determinant for UL33-G_s signaling, and the edge interactions between ICLs and the globular domain of G_s are supportive, consistent with the observed structural flexibility between UL33 and G_s. " Again, I don't see how it follows that these interactions are consistent with observed structural flexibility.

Thank you for pointing this out. Our interpretation is based on the comparison with typical human GPCRs (e.g., chemokine receptors), which often exhibit extensive interactions between ICL2/ICL3 and the G protein, thereby stabilizing the interface and limiting global flexibility. In contrast, the UL33-G_s interface appears less extensive in the loop regions. Therefore, the functional finding that the core $\alpha 5$ interaction is the critical determinant, while ICL interactions are merely supportive, is consistent with the higher degree of global flexibility observed between UL33 and G_s in the structural analysis.

Lines 277-279." Indeed, our cryo-EM analysis suggested structural flexibility between UL33 and the G proteins (Fig. S1; Fig. S2). Despite this, UL33 has acquired the ability to promiscuously couple to G_s, G_i, and G_{q3}." Why "despite"? wouldn't more flexibility allow a receptor to accommodate a larger number of G proteins? I also do not see how Figures S1 and S2 necessarily indicate high flexibility.

Again, we apologize for the unclear words.

1) We removed the sentence "Indeed, our cryo-EM analysis suggested structural flexibility between UL33 and the G proteins (Fig. S1; Fig. S2)."

This is intended to explain the outcome of the narrow intracellular cavity, but we agree that it is rather confusing.

2) We corrected the following sentence to "Despite the narrow intracellular cavity with inwardly retained TM6, ..." to be clear.

Our interpretation is that UL33's G protein promiscuity is derived from the shallow insertion of G α - $\alpha 5$ instead of deep insertion observed in typical human GPCRs, likely at the expense of maximal signaling efficiency, and the flexibilities between UL33 and G proteins are the outcome of this tenuous interaction.

We also think that, in general, flexibility or the more dynamic nature of TM6 movement might contribute to the G protein promiscuity but in a distinct mechanism, although it should be case-by-case.

Final paragraph. I do not really follow the logic. Molecules that would bind here may more likely be PAMs than NAMs. I would predict most molecules that bind in there would keep the N-terminal methionine in its pocket and thus keep on activating the receptor. Seems like one would be better off finding compounds that displace the N-terminus because based on the density, it is not bound very tightly. In the end, the "allosteric modulator" aspect of the paper is not a strong selling point to me and the authors may want to tone that down.

We appreciate the reviewer's critical perspective, and apologize for the misleading terminology: the molecules we envisioned would target this site via the tunnel and affect both the empty pocket and the adjacent N-terminal peptide. We agree with the reviewer that we cannot predict whether a molecule targeting the empty site would function positively or negatively on signaling, as it might either disrupt or stabilize the authentic active conformation. Furthermore, the idea of developing a molecule that completely displaces the N-terminal peptide would be an attractive, alternative approach, leveraging the likely dynamic nature of the N-terminal peptide. Thus, we have significantly toned down the statements and revised the section to offer both possibilities and present a more balanced discussion on therapeutic strategies, as outlined in the response to your earlier comment (Major Issue #2).

Question: Although it clearly worked, the use of a fusion to the Gbeta1 C-terminus seems like a risky move given that the C-terminal carboxylate of the protein forms structural interactions with the protein core. Is there any chance this unusual nanobit linkage might perturb the overall configuration between the G protein and the receptor? Also, the authors might want to explain why they went with Nanobit in the first place. Is it because they could not isolate a complex without it (i.e., did they try)?

Thank you for this question regarding the NanoBiT tethering strategy. To clarify, the HiBiT tag is fused to the C-terminus of the G β subunit, not G α , via a 15-amino acid flexible linker, which is distal to the core receptor-G protein interface (Response Figure 1). This tethering approach is an established method for preparing nucleotide-free GPCR-G protein complexes (Duan *et al.*, *Nat Commun*, 2020).

Response Figure 1: Schematic of the NanoBiT tethering (dark gray: linker, Nb32 not shown).

We did not test non-tethered coupling but utilized this approach because we hypothesized that the interaction might be tenuous, and tethering would enhance complex stability. Additionally, the strategy integrates well with our optimized one-pot co-expression system using inducible Expi293F cells.

Reviewer #1 (Remarks to the Author):

In their revised manuscript, the authors have comprehensively responded to all points raised. The revisions, which satisfactorily address the reviewers' comments, have been incorporated and have strengthened the work. It could be considered for publication.

We thank you again for your constructive comments, which have helped us to improve the manuscript. We appreciate your recommendation for publication.

Reviewer #2 (Remarks to the Author):

I would like to thank the authors for addressing my comments from the initial review of their manuscript. I am satisfied with their answers to not only my questions but also the other reviewers' questions and comments and would like to recommend the manuscript for acceptance.

We thank you for reviewing our revised manuscript and for your positive feedback. We appreciate your recommendation for acceptance.

Reviewer #3 (Remarks to the Author):

The authors have satisfactorily addressed my comments. By the way, I did write for the last comment that the nano bit linkage in question was to the C-terminus of Gbeta in my comment to the authors (not Galpha as the rebuttal says). However, given that this region is relatively distant from the receptor interface I don't think it is a major issue. It's a strange choice, but I guess it works.

We thank you for finding our revisions satisfactory. We apologize for the misunderstanding regarding the NanoBiT linkage site in our previous response. We agree with your assessment that the C-terminus of G β is located at the periphery of the complex, away from the receptor interface. As shown in Response Fig. A, there is sufficient space to accommodate the fusion via a flexible Gly-based linker.

Response Fig. A: Structure of the G β C-terminal region without the extended linker (PDB: 7JHJ).